# Interplay between metavalent bonds and dopant orbitals enables the design of SnTe thermoelectrics

Guodong Tang [1,10] ✉, Yuqi Liu[1,10], Xiaoyu Yang[2,10], Yongsheng Zhang [3], Pengfei Nan [2], Pan Ying [1], Yaru Gong[1], Xuemei Zhang[4,5] ✉, Binghui Ge [2], Nan Lin[6], Xuefei Miao[1], Kun Song [7], Carl-Friedrich Schön[6], Matteo Cagnoni [8], Dasol Kim[6], Yuan Yu [6] ✉ & Matthias Wuttig [6,9] ✉

Engineering the electronic band structures upon doping is crucial to improve the thermoelectric performance of materials. Understanding how dopants influence the electronic states near the Fermi level is thus a prerequisite to precisely tune band structures. Here, we demonstrate that the Sn-s states in SnTe contribute to the density of states at the top of the valence band. This is a consequence of the half-filled p-p σ-bond (metavalent bonding) and its resulting symmetry of the orbital phases at the valence band maximum (L point of the Brillouin zone). This insight provides a recipe for identifying superior dopants. The overlap between the dopant s- and the Te p-state is maximized, if the spatial overlap of both orbitals is maximized and their energetic difference is minimized. This simple design rule has enabled us to screen out Al as a very efficient dopant to enhance the local density of states for SnTe. In conjunction with doping Sb to tune the carrier concentration and alloying with $AgBiTe_2$ to promote band convergence, as well as introducing dislocations to impede phonon propagation, a record-high average $ZT$ of 1.15 between 300 and 873 K and a large $ZT$ of 0.36 at 300 K is achieved in $Sn_{0.8}Al_{0.08}Sb_{0.15}Te$-4%$AgBiTe_2$.

Thermoelectric materials enable the direct conversion between heat and electricity, emerging as potential candidates to alleviate energy problems[1,2]. The conversion efficiency of thermoelectric devices can be judged by the figure of merit, defined as $ZT = (S^2\sigma/\kappa_T)T$, where $S$, $\sigma$, $\kappa_T$ and $T$ are the Seebeck coefficient, the electrical conductivity, the total thermal conductivity, and the absolute temperature, respectively[3]. $\kappa_T$ consists of the electronic thermal conductivity ($\kappa_e$), the lattice thermal conductivity ($\kappa_L$), and the bipolar thermal conductivity ($\kappa_B$) for low-bandgap compounds[4,5]. Historically, thermoelectric power generation heavily relies on PbTe, which has been used

[1]National Key Laboratory of Advanced Casting Technologies, MIIT Key Laboratory of Advanced Metallic and Intermetallic Materials Technology, Engineering Research Center of Materials Behavior and Design, Ministry of Education, Nanjing University of Science and Technology, Nanjing 210094, China. [2]Key Laboratory of Structure and Functional Regulation of Hybrid Materials of Ministry of Education, Institutes of Physical Science and Information Technology, Anhui University, Hefei 230601, China. [3]Advanced Research Institute of Multidisciplinary Sciences, Qufu Normal University, Qufu, Shandong Province 273165, China. [4]School of Physics and Electronic Information Engineering, Engineering Research Center of Nanostructure and Functional Materials, Ningxia Normal University, Guyuan, Ningxia 756000, China. [5]Science Island Branch of Graduate School, University of Science and Technology of China, Hefei 230026, China. [6]Institute of Physics (IA), RWTH Aachen University, 52056 Aachen, Germany. [7]School of Mechanical and Power Engineering, Nanjing Tech University, 30 Puzhu South Road, Nanjing, Jiangsu 211816, China. [8]Department of Electronics and Telecommunications, Politecnico di Torino, Corso Duca degli Abruzzi 24, 10129 Torino, Italy. [9]Peter Grünberg Institute—JARA-Institute Energy-Efficient Information Technology (PGI-10), Forschungszentrum Jülich GmbH, Jülich 52428, Germany. [10]These authors contributed equally: Guodong Tang, Yuqi Liu, Xiaoyu Yang. ✉e-mail: tangguodong@njust.edu.cn; saxuemei@mail.ustc.edu.cn; yu@physik.rwth-aachen.de; wuttig@physik.rwth-aachen.de

in space exploration missions since the 1960s by NASA[6]. Yet, the environmentally hazardous element Pb is creating pollution problems. Developing Pb-free materials has thus become an imperative in many areas including thermoelectrics[7–10].

As a Pb-free analog of PbTe, SnTe shows a number of similarities to PbTe, including the rocksalt crystal structure, the high electrical conductivity, the small bandgap, and the low band effective mass, as well as the strong lattice anharmonicity[11]. These similarities for PbTe and SnTe stem from a particular chemical bonding mechanism that both share. It is called metavalent bonding, which describes a special bonding configuration of "two center-one electrons (2c–1e)", distinctively different from conventional covalent, ionic and metallic bonding[12–14]. In this regard, SnTe should also show comparable thermoelectric performance to PbTe[15–20]. However, the much lower formation energy of cation vacancies in SnTe than that in PbTe[21] leads to an intrinsic hole carrier concentration on the order of $10^{21}\,cm^{-3}$ for SnTe[22]. Optimizing the carrier concentration is often the first and foremost step to enhance thermoelectric performance. The optimum carrier concentration ($n_{opt}$) depends on the density-of-states (DOS) effective mass ($m^*$) and temperature as $n_{opt} \propto (m^*T)^{1.5}$ [23]. It generally lies in the range of $10^{19}-10^{21}\,cm^{-3}$ depending on the electronic band structure, which affects $m^*$ [24]. For heavy-band thermoelectrics (large $m^*$) such as Fe(VNb)Sb half-Heusler alloys ($m^*$= 2.5 $m_e$), it is very difficult to achieve the high optimum carrier concentration due to the limited solubility of dopants[25]. Alternatively, reducing $m^*$ by band sharpening (increasing band dispersion) can optimize the carrier concentration at a lower doping level to improve performance[25]. In stark contrast, SnTe is a light-band semiconductor (small $m^*$=0.13 $m_e$), which has a low $n_{opt}$ of about $10^{19}\,cm^{-3}$ [26]. Yet, this value is about 1-2 orders of magnitude lower than the observed carrier concentration in pristine SnTe. Thus, decreasing the carrier concentration in SnTe becomes critical for ZT enhancement. Even though Sn self-compensation and other doping strategies can lower the carrier concentration to some extent, the value typically obtained is still considerably larger than the optimum value, leading to very poor ZT values at low to intermediate temperatures[27,28]. Moreover, a low carrier concentration concomitantly leads to a small electrical conductivity. It thus appears that the $n_{opt}$ can be realized in light-band thermoelectrics by increasing the $m^*$ rather than significantly decreasing the carrier concentration of the material.

The key question becomes how to significantly increase the density-of-states effective mass. The DOS effective mass depends on the details of the band structure at the Fermi level[29]. The band convergence effect, i.e., moving multiple electronic bands closer in energy to provide more carrier pockets[30], is an effective way to enhance $m^*$. Since SnTe shows a large energy offset (0.3–0.4 eV) between the light-hole band (L band) and a heavy-hole band (Σ band), it is more difficult to realize the same level of band convergence that has been achieved in PbTe[4]. Nevertheless, some dopants such as Mg[31] Ca[32], Cd[33], Hg[34], and Mn[35] were reported to enable a band convergence effect in SnTe. Yet, these elements also form precipitates in SnTe. A stronger band convergence requires a high solubility of dopants. This has been achieved by alloying SnTe with other metavalently bonded compounds such as AgSbTe$_2$[36] but more candidates such as AgBiTe$_2$ are available in a chemical bonding map[19,37,38], increasing the diversity of material choices to achieve band convergence. Various dopants and their impact on the electronic structure are summarized in Table S1.

Besides band convergence, it has been demonstrated that $m^*$ can also be enhanced by introducing 'resonant' levels to create a DOS hump near the Fermi energy. This effect can be realized by introducing elements such as V[39], W[40], Mo[41], Zn[42,43] and In[44] in SnTe. A similar phenomenon can also be achieved in Tl-doped PbTe[45]. This hump in DOS is due to the strong hybridization between the Tl-s/In-s and the Te-p states. However, it is difficult to understand how the Tl-s/In-s states can 'resonate' with the valence band of PbTe/SnTe at the Fermi

level given the large difference in energy of these atomic orbitals. Moreover, a rational design rule for dopants to maximize the local DOS near the Fermi level seems still obscure even though some conditions for DOS enhancement have been discussed by Heremans et al.[45].

In this work, we unravel the impact of the dopant s-state on the valence band of SnTe near the Fermi level. The special bonding in SnTe (metavalent bonding) and the resulting symmetry of corresponding orbital phases at the Γ and L points are key factors as shown in the following. Understanding the impact of orbital overlap helps us to identify superior dopants. Al is identified as an effective dopant, which increases the local DOS of SnTe near the Fermi level due to the pronounced spatial and energetic overlap between dopant s- and Te p-orbitals. We finally realize a high ZT of 0.36 at 300 K and a record-high average ZT of 1.15 in the range of 300-873 K in Sn$_{0.8}$Al$_{0.08}$Sb$_{0.15}$Te-4%AgBiTe$_2$. This work reveals the underlying mechanisms of the IV-s states for the electrical properties in IV-VI compounds and provides a clear design rule to tailor the local DOS near the Fermi level.

## Results

### Screening dopants to tailor the density of states near the Fermi level

To investigate the impact of dopants on the electrical transport properties of thermoelectric materials like SnTe, we need to first understand the electronic states of SnTe at the Fermi level. Figure 1a shows the orbital and element-resolved density-of-states (DOS) for pristine SnTe. It can be seen that the conduction band and valence band are dominated by the Sn-p and Te-p states, respectively, with additional pronounced contributions from the p-p orbital overlap, leading to the large band dispersion. The atomic energy levels of Sn-s and Te-s orbitals are much lower than the valence band. Thus, these s-orbitals have much less overlap with other orbitals. However, there is a small Sn-s contribution right below the Fermi level and a small Te-s contribution right above the Fermi level. To better understand the origin of these features, we plotted the orbital projected band structures, so-called fat bands, as shown in Fig. 1b and Fig. S1. There the wave function corresponding to each band is projected onto the relevant orbitals. In the vicinity of the Fermi levels, these orbitals are the Sn-5s and 5p states, as well as the Te-5s and 5p states. We find that the contribution of Sn-s state close to the Fermi level in Fig. 1a is pronounced at the L-point of the Brillouin zone, but is absent at the Γ point. This is similar to the Te-s state, which is also absent at the Γ point but prevalent at the L point. This phenomenon is very striking since the energy difference between Sn-s and Te-p at the Γ point is much smaller than that at the L point, as demonstrated in Fig. 1b. From the perspective of perturbation theory by considering the energy difference between orbitals, one would thus have expected a larger contribution at the Γ point than the L point. Yet, this is not in line with the calculated results.

Figure 1c sketches a combination of the electronic band structure (physical perspective) and the phase of orbitals for different states (chemical perspective) at the Γ and L points in the first Brillouin zone. Here, the phase of orbitals refers to the mathematical sign and behavior of the wavefunction of an electron in the orbital. Based on the Bloch theorem, the phase of a certain atomic orbital is the same, i.e., in phase, as it repeats in other unit cells at the Γ point[46,47]. Thus, Sn-p and Te-p orbitals form a bonding configuration (the same phase orbitals are head-to-head) and lower the energy of the valence band at the Γ point. Different from the wavefunction of p-orbital which has a shape depicted as lobes and has a nodal plane, the wavefunction of s-orbital is spherical. As a consequence, the Sn-s states do not contribute to the bonding at the Γ point since bonding and anti-bonding configurations between Sn-s and Te-p orbitals cancel each other. On the contrary, at the L point, the adjacent Te-p orbitals are out-of-phase (the same for Sn-p orbitals) since the wavelength is two times the unit cell size. This leads to a non-bonding configuration (different phase orbitals are

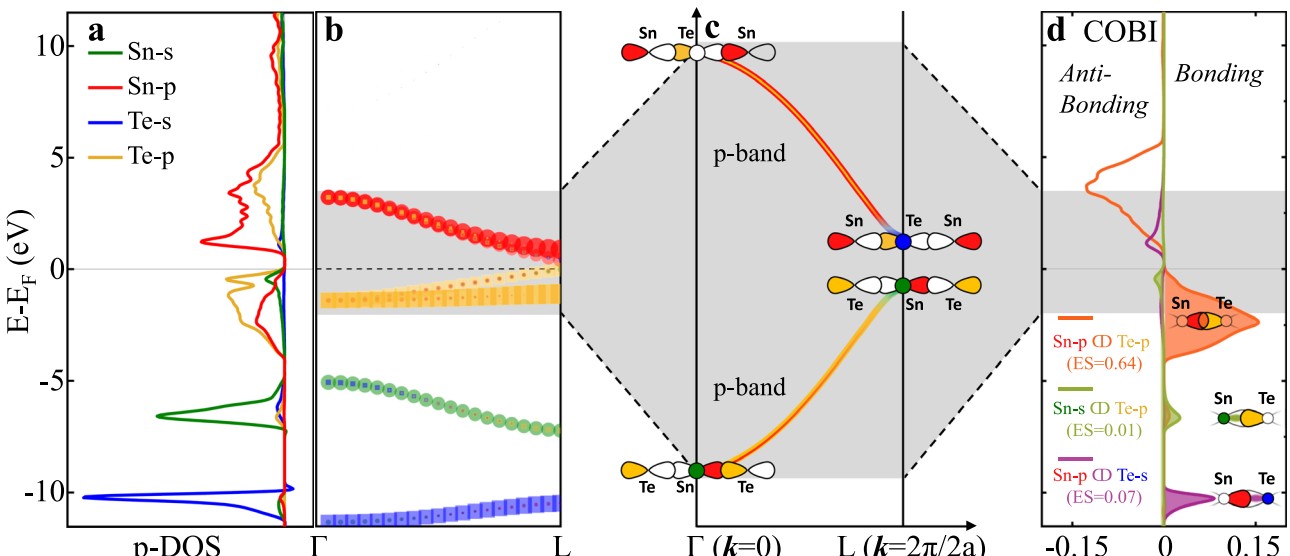

**Fig. 1 | Electronic band structures and chemical bonds in SnTe. a** Elemental and orbital resolved partial density-of-states (p-DOS). A small hump of Sn-s state is observed at the valence band maximum. **b** Orbital-projected electronic band structures between the Γ and L points. **c** Schematic of the conduction and valence bands between the Γ and L points. The dumbbells and circles represent the p-orbitals and s-orbitals, respectively. The filled pattern and open pattern represent the orbital phase of "+" and "−", respectively. The same phase orbitals combine to form bonding states while the different phase orbitals combine to form anti-bonding states. **d** Crystal orbital bond index (COBI) analyses indicate that Sn-p and Te-p orbital overlap dominates the formation of bonds. In contrast, the bond order of Sn-s and Te-p overlap is zero, verifying that there is no bond formation.

combined) between Sn-p and Te-p orbitals. However, this out-of-phase feature for Te-p orbitals naturally allows contributions from the Sn-s states at the L point due to the spherical symmetry of s-orbitals. The Sn-s and Te-p orbital overlap forms an anti-bonding state, which increases the energy level of the valence band at the L point. Therefore, the anti-bonding configuration between Sn-s and Te-p at the L point contributes to the valence band maximum at the L point, where the Fermi energy level is located. This is also why the Sn-s states are important for the electrical transport properties in SnTe. Nevertheless, the valence band is still dominated by the p-p overlap as revealed by the DOS and fat bands.

The crystal orbital bond index (COBI) in Fig. 1d quantifies the contributions of different orbital overlaps to bonding. Apparently, the p-p orbital overlap dominates bond formation. In contrast, the anti-bonding states and bonding states between Sn-s and Te-p cancel each other, resulting in a net electron sharing value of 0.01, which means a bond order of about zero[48]. This indicates that the orbital overlap between Sn-s and Te-p does not contribute to bonding. This is no surprise from simple arguments of the formation of molecular orbitals. The Sn-s state is completely filled and it interacts with the state derived from the Te p-state, which is also predominantly filled. Hence, the overlap of two filled states does not contribute to bonding; an argument, which is also used to explain the instability of the He₂ molecule. On the other hand, this argument also explains why there is a slightly larger, yet still small contribution to bond formation from the Sn p- to Te s- orbital. This is surprising at first sight since the energetic overlap of these two orbitals is significantly smaller than for the Te p- to Sn s- orbital. Yet, a fraction of the Sn p-state is not fully occupied. Hence, the bonding part and the antibonding part of the Te s- and Sn p-state are not both equally filled, as can be seen in Fig. 1d. Therefore, we can conclude that the Sn-s orbital is important for the electrical transport properties because it contributes to the states near the Fermi level. Yet, it is not relevant for bond formation. Instead, the bonding is dominated by the p-states. Since SnTe has in total 6 p-electrons and an effective coordination number of 6, the average number of p-electrons for each bond is just one. This is different from the electron pair required for a typical covalent bond described by Lewis[49]. This special

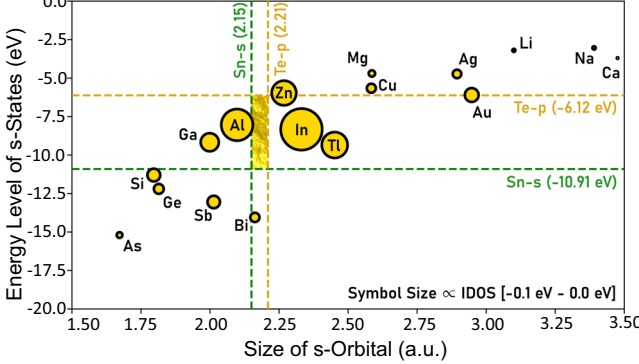

**Fig. 2 | Design rule for dopants in terms of spatial and energy overlaps between orbitals.** A plot spanned by the size of s-orbitals and the energy level of s-states for dopants. The corresponding size and energy of Sn-5s and Te-5p orbitals are also indicated in the map. The "golden rectangle" area highlights the "sweet spot" for efficient dopants to create a large DOS 'hump' in SnTe, where Al and In are indicated as the most efficient candidates as demonstrated by their symbol sizes.

bonding mechanism, where the p-states are only half-filled (2c−1e configuration) is called metavalent bonding[12,14,50,51]. The resulting symmetry in the phase of orbitals allows the involvement of Sn-s states to charge carrier transport, but not to bonding.

Based on the above understanding, we can now tune the local DOS near the Fermi level (L point) by manipulating the overlap between the s-states of dopants that substitute Sn and Te-p states. The degree of overlap between orbitals depends on both their spatial size and energy differences. We have calculated the atomic energy levels of many dopant atoms (Table S2) using the full-potential full-electron Wien2K code[52]. Figure 2 plots the size of s-orbitals in atomic units (Bohr radius)[53] and the energy level of s-states of various dopants, where the corresponding values for Sn-s and Te-p orbitals are marked by dashed lines. It is expected that the smaller difference in the size and energy between dopant-s and Te-p states, the larger the orbital

overlap between these states, and the stronger the DOS contribution ('hump') at the Fermi level. The "golden rectangle" area highlights the "sweet spot" for the most efficient dopants to create a DOS 'hump' in SnTe because the orbital overlap is expected to be maximized in this area based on the argument of spatial and energetic overlap. The integrated partial DOS (iDOS) of SnTe with these different dopants is shown in Fig. S2. The symbol size in Fig. 2 scales to the difference in iDOS between 0 eV and -0.1 eV. Here, the Fermi energy level is defined at 0 eV in the DFT calculations. The cutoff energy of -0.1 eV is chosen based on the fact that only the states near a few $k_BT$ of the Fermi energy can contribute to the charge transport. Figure S2 also indicates that a small change in the energy range starting from the Fermi energy level to a few $k_BT$ will not change the value of iDOS significantly. This supports the robustness of the relative trend of the symbol size in Fig. 2. We find that doping Al in SnTe can significantly increase the local DOS near the Fermi level, which is induced by the Al-s states as illustrated in Fig. S3a. This state has been sometimes described as a "resonant level", for example, in In-doped or Zn-doped SnTe, which has been theoretically and experimentally verified[22,42,43]. This further corroborates the usefulness of Fig. 2 in identifying effective dopants to enhance DOS. Yet, the wording of "resonant level" potentially masks the origin of this state. The atomic energy of Al-s/In-s states is far below the valence band, which thus cannot resonate with the Fermi level. The reason why the Al-s states appear at the L point has been ascribed to the combination of half-filled p-p σ-bond (metavalent bonding)[12] and the phase of different orbitals, as explained above. In contrast to Al, the atomic energy of Sb-s states is even lower than that of Sn-s states. This results in a weaker overlap between Sb-s and Te-p and thus small contributions at the Fermi level, as illustrated in Fig. S3b. The explanation for the dopant s-state contribution to the DOS at the Fermi level ('hump') thus provides a simple recipe for how to identify the most effective dopant, one only needs to maximize the spatial and energetic overlap of the two orbitals involved (here: dopant s- and Te p-state).

## Crystal structure

Based on the screened dopants, we have prepared a series of $Sn_{1.03-x-y}Al_xSb_yTe$ and $Sn_{0.8}Al_{0.08}Sb_{0.15}Te-z\%AgBiTe_2$ samples. Sb is a typical dopant to tune the carrier concentration of SnTe[54,55]. $AgBiTe_2$ is used to introduce band convergence and to lower the lattice thermal conductivity, as will be discussed later. Powder X-ray diffraction (XRD) patterns of these samples (Fig. 3a) can be well indexed to rock-salt SnTe (PDF #46-1210, Fm-3m space group). The lattice parameters, determined upon XRD refinement (one example is demonstrated in Fig. 3b), decrease with increasing Al and Sb codoping levels and $AgBiTe_2$ content (Fig. S4). This can be primarily attributed to the smaller radius of $Al^{3+}$ (0.54 Å) and $Sb^{3+}$ (0.76 Å) in contrast to $Sn^{2+}$ (0.93 Å), as well as the smaller lattice parameter of $AgBiTe_2$ (6.155 Å) compared to $Sn_{0.8}Al_{0.08}Sb_{0.15}Te$ (6.293 Å)[56]. Note that the lattice parameters should in principle also be influenced by the bonding and anti-bonding states of corresponding molecular orbitals according to a phenomenological bond length – bond strength rule. Short bonds are typically stronger bonds, while longer bonds are typically weaker bonds. Normally, electrons in the bonding orbitals stabilize the molecule, pulling the nuclei closer and shortening the bond length. On the contrary, electrons in the anti-bonding states can destabilize the molecule by exerting a repulsive force between the nuclei. Thus, the bond length increases with increasing the population of occupied anti-bonding states. However, in this work, the occupied anti-bonding states are created by the overlap between Sn-s/dopant-s and Te-p orbitals. The destructive effect of these anti-bonding states is compensated for by the equivalent constructive effect of the bonding states formed by these orbitals. This has been proven and discussed in Fig. 1d. Thus, the anti-bonding state in SnTe has a negligible effect on the bond length. No impurities can be observed within the detection limit of XRD (Fig. 3a) and scanning electron microscopy (SEM) attached with energy dispersive spectroscopy (EDS) (Fig. S5c-h). High-angle annular dark-field scanning transmission electron microscopy

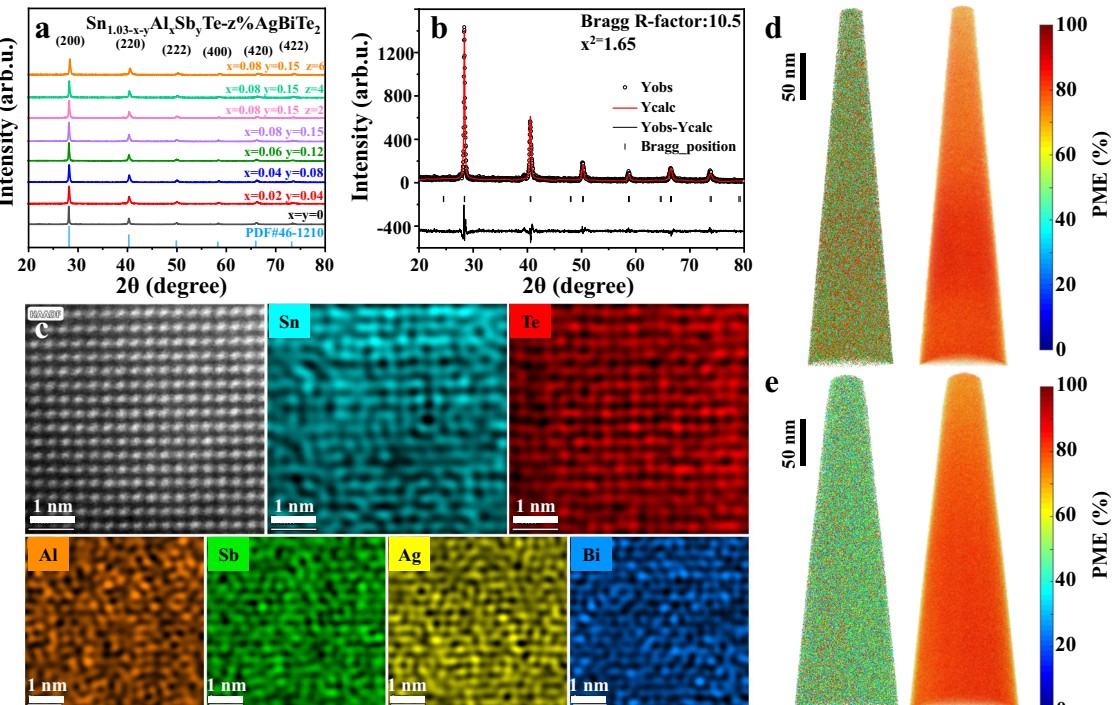

**Fig. 3 | Structural characterization and chemical bonding mechanism of the SnTe-based samples studied. a** Powder XRD patterns of $Sn_{1.03-x-y}Al_xSb_yTe-z\%AgBiTe_2$ (x = 0, 0.02, 0.04, 0.06, 0.08, y = 0, 0.04, 0.08, 0.12, 0.15, z = 2, 4, 6), **b** Rietveld refinement details of the XRD pattern of $Sn_{0.8}Al_{0.08}Sb_{0.15}Te-4\%AgBiTe_2$ sample. **c** Atomic-resolution HAADF-STEM image and corresponding EDS mapping results of the same area. **d** APT reconstruction of sample $Sn_{0.8}Al_{0.08}Sb_{0.15}Te$ and corresponding PME map. **e** APT reconstruction of sample $Sn_{0.8}Al_{0.08}Sb_{0.15}Te-4\%AgBiTe_2$ and corresponding PME map. The color scales represent different PME values from 0% to 100% with critical numbers labeled.

(HAADF-STEM) image and corresponding mappings (Fig. 3c) of $Sn_{0.8}Al_{0.08}Sb_{0.15}Te-4\%AgBiTe_2$ clearly show elemental distributions of Ag, Bi, Sb, and Al. All these characterizations indicate a high solubility of dopants in SnTe with compositions close to their nominal stoichiometry. This large miscibility between $Sn_{0.8}Al_{0.08}Sb_{0.15}Te$ and $AgBiTe_2$ can be attributed to the metavalent bonding mechanism[36] for both compounds. A similar phenomenon has also been observed in $SnTe-AgSbTe_2$ alloys[57]. The metavalent bonding mechanism in the present compounds has been confirmed by a high value for the "probability of multiple events (PME)"[58] obtained in atom probe tomography (APT) measurements, as presented in Fig. 3d–e. The PME value represents the probability of creating more than one fragment under the illumination of one successful laser pulse in APT. In general, a PME value larger than 60% is a hallmark of metavalently bonded compounds[58]. The large solubility of dopants facilitated by metavalent bonding is a prerequisite to effectively tune the electronic energy band structures, which influences the transport properties as elaborated below.

## Electrical transport properties

Figure 4a presents the electrical conductivity ($\sigma$) as a function of temperature for $Sn_{1.03-x-y}Al_xSb_yTe-z\%AgBiTe_2$ samples. Compared with pure SnTe and Sn self-compensated $Sn_{1.03}Te$, $\sigma$ for Al and Sb codoped samples decreases with increasing Al and Sb doping levels due to the reduction in both carrier concentration ($n_H$) and carrier mobility ($\mu_H$) (Fig. 4b). The reduction of carrier concentration can be ascribed to the substitution of $Sb^{3+}$ and $Al^{3+}$ at the $Sn^{2+}$ site which supplies extra electrons to compensate for the high hole carrier concentration. Due to enhanced scattering of charge carriers by substitutional point defects, $Sn_{1.03-x-y}Al_xSb_yTe$ samples possess lower $\mu_H$ than that of pristine $Sn_{1.03}Te$ (Fig. 4b). After alloying $Sn_{1.03-x-y}Al_xSb_yTe$ with $AgBiTe_2$, $\sigma$ is further decreased due to the reduction of $n_H$. However, alloying $AgBiTe_2$ does not significantly decrease the $\mu_H$ although the number of point defects increases (Fig. 4b). This is mainly because Ag and Bi can fill Sn vacancies to weaken the vacancy scattering of charge carriers thus maintaining a high carrier mobility[57]. Figure 4c shows the temperature dependence of the Seebeck

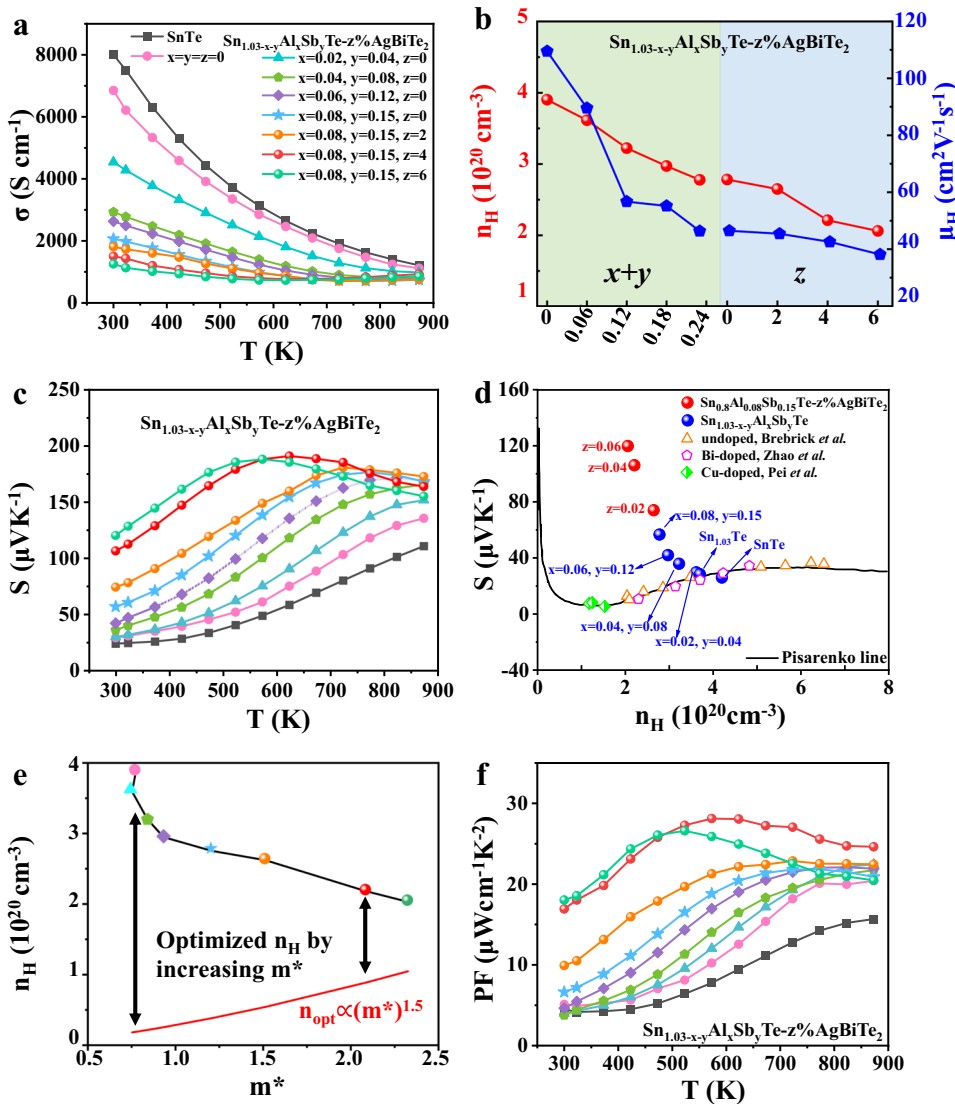

**Fig. 4 | Electrical transport properties of $Sn_{1.03-x-y}Al_xSb_yTe-z\%AgBiTe_2$ samples.** **a** Temperature-dependent electrical conductivity. **b** Compositional dependence of the carrier concentration ($n_H$) and the carrier mobility ($\mu_H$) at room temperature. **c** Temperature-dependent Seebeck coefficients. **d** Room temperature Seebeck coefficients as a function of the carrier concentration of $Sn_{1.03-x-y}Al_xSb_yTe-z\%$ $AgBiTe_2$ samples. For comparison, previously reported data of undoped SnTe, Cu-doped SnTe, and Bi-doped samples as well as theoretical Pisarenko curve based on a two-valence-band model are given. **e** Optimized $n_H$ with increasing $m^*$ of $Sn_{1.03-x-y}Al_xSb_yTe-z\%AgBiTe_2$. The data points share the same legend in Fig. 3a. **f** Power factor of $Sn_{1.03-x-y}Al_xSb_yTe-z\%AgBiTe_2$ samples.

coefficient ($S$) for $Sn_{1.03-x-y}Al_xSb_yTe-z\%AgBiTe_2$ samples. The $S$ values increase with increasing the content of Al and Sb as well as $AgBiTe_2$, consistent with the decreased carrier concentration illustrated in Fig. 4b. Yet, the decreased $n_H$ is not solely responsible for the enhanced $S$. Figure 4d depicts the room-temperature $S$ of $Sn_{1.03-x-y}Al_xSb_yTe-z\%AgBiTe_2$ samples and other typical SnTe-based compounds with various dopants[59–61] as a function of $n_H$. The solid line is the Pisarenko relation between $S$ and $n_H$ calculated based on a two-parabolic band model[62]. The experimental data of undoped SnTe[59], Bi-doped[60], and Cu-doped SnTe[61] fall on the theoretical Pisarenko line[22]. In contrast, $S$ of $Sn_{1.03-x-y}Al_xSb_yTe-z\%AgBiTe_2$ samples exhibit higher values than the theoretical Pisarenko line, which is indicative of modifications in the electronic band structure. We have previously demonstrated that Sb doping has a relatively weak impact on the electronic band structures of SnTe[36], which has also been shown and interpreted in Fig. 2 and Fig. S6. Tan et al.[63] have also demonstrated that doping Sb in SnTe can induce a very weak band convergence effect yet could not be observed experimentally. Therefore, the significant enhancement of $S$ can be mainly attributed to band modification created by Al doping and $AgBiTe_2$ alloying. We have explained why and how Al can create extra DOS near the Fermi level in SnTe in Fig. 1 and Fig. S7a, b. The introduction of Sb in Al-doped SnTe does not significantly change the valence band structure. Instead, Sb mainly influences the conduction band with a strong contribution from the Sb-p orbital (Fig. S7c, d). Further alloying $AgBiTe_2$ can not only retain the contribution from Al but also lower the energy offset between the L and $\Sigma$ points ($\Delta E(L-\Sigma)$) (Fig. S7e, f). To identify the effect of $AgBiTe_2$ alloying on the electronic band structures, we have also calculated the band structure of $Sn_{0.926}Ag_{0.037}Bi_{0.037}Te$ alloy. As shown in Fig. S8, Ag and Bi codoping leads to a decrease in the band offset of $\Delta E(L-\Sigma)$ from 0.35 eV for the pristine SnTe to 0.19 eV for the $Sn_{0.926}Ag_{0.037}Bi_{0.037}Te$ alloy, which is favorable for a strong band convergence at a suitable doping level. Such band convergence can also increase the density of states around the VBM in addition to the extra DOS hump induced by Al (Fig. 1).

The modification of the band structures will directly impact the density-of-states effective mass ($m^*$), which can be estimated using $n_H$ and $S$ as proposed by Snyder et al.[64] derived from the simple free electron model using a constant mean-free path.

$$\frac{m^*}{m_e} = 0.924 \left(\frac{300\,K}{T}\right)\left(\frac{n_H}{10^{20}cm^{-3}}\right)^{2/3}$$
$$\left[\frac{3\left(exp\left[\frac{|S|}{\frac{k_B}{e}}-2\right]-0.17\right)^{2/3}}{1+exp\left[-5\left(\frac{|S|}{\frac{k_B}{e}}-\frac{k_B}{|S|}\right)\right]}+\frac{\frac{|S|}{\frac{k_B}{e}}}{1+exp\left[5\left(\frac{|S|}{\frac{k_B}{e}}-\frac{k_B}{|S|}\right)\right]}\right] \quad (1)$$

where $k_B$ is the Boltzmann constant, $m_e$ is the electron mass, and $e$ is the electronic charge. Subtle changes in the band structure are easily seen in $m^*$. As claimed by Snyder et al.[64], small variations in $m^*$ can be used to identify unusual phenomena in materials. In special cases, dopants that produce distortions in the density of states can be interpreted as modifications of the single band effective mass and are measured as a shift in $m^*$, as observed in p-type Tl-doped PbTe[65]. In this regard, the calculated $m^*$ can be used to qualitatively predict the relationship between $n_{opt}$ and $m^*$ even though the band structure is not parabolic due to the extra DOS hump induced by Al. Figure 4e plots the calculated $m^*$ and the measured $n_H$ at room temperature. It shows that $m^*$ increases with increasing content of Al and $AgBiTe_2$. This significantly reduces the gap between the experimental $n_H$ and the theoretical optimum carrier concentration since $n_{opt}$ increases with $m^*$ as a power law with an exponent of 1.5[23], as delineated by the solid line. Generally, optimizing the carrier concentration is often the foremost step to improve the thermoelectric properties. For materials with an

intrinsically high concentration of vacancies and small band effective masses such as SnTe, it is very difficult to optimize the carrier concentration by reducing the number of charge carriers. Alternatively, increasing the value of $m^*$ could be another approach to reach $n_{opt}$ while keeping a relatively high value of practical carrier concentration. As shown in Fig. 4e, after Al doping, the $m^*$ increases from 0.76 $m_e$ to 1.2 $m_e$ for $Sn_{1.03}Te$ and $Sn_{0.8}Al_{0.08}Sb_{0.15}Te$, respectively. The $m^*$ further increases from 1.2 $m_e$ to 2.08 $m_e$ after alloying with $AgBiTe_2$. As a consequence, the optimum carrier concentration required is significantly increased, which is closer to the practical charge carrier density. The relatively optimized carrier concentration leads to a dramatic enhancement of the power factor, especially at low to intermediate temperatures, as shown in Fig. 4f. Power factor increases from 20.47 $\mu W\,cm^{-1}\,K^{-2}$ for $Sn_{1.03}Te$ ($m^* = 0.76$ $m_e$) to 28.1 $\mu W\,cm^{-1}\,K^{-2}$ for the $Sn_{0.8}Al_{0.08}Sb_{0.15}Te-4\%AgBiTe_2$ ($m^* = 2.08$ $m_e$) sample. The decrease of PF for samples with a high content of $AgBiTe_2$ ($z = 4$ and 6) at elevated temperatures could be due to the bipolar effect induced by intrinsic excitation. The onset temperature of bipolar conduction depends on both the electronic bandgap and the concentration of majority carriers. We measured the bandgap of $Sn_{1.03-x-y}Al_xSb_yTe-z\%AgBiTe_2$ samples experimentally by UV−Vis−NIR absorption spectrum, as listed in Table S3. The reasons for the evolution of bandgaps are also discussed in the supporting information. Figure 4c indicates that the bipolar effect is prominent at high temperatures only in samples with a high content of $AgBiTe_2$. The decreased bandgap is partly responsible for this behavior, while the reduced carrier concentration (Fig. 4e) is another reason for the enhanced bipolar conduction.

## Thermal transport properties

Minimization of the lattice thermal conductivity in SnTe is another imperative for enhancing thermoelectric performance. The total thermal conductivity ($\kappa_T$) of the Al and Sb codoped samples is considerably lower than that of pure SnTe and Sn self-compensated $Sn_{1.03}Te$. $\kappa_T$ of the samples alloyed with $AgBiTe_2$ is further suppressed compared to the $AgBiTe_2$-free sample (Fig. 5a). The lattice thermal conductivity ($\kappa_L$) was obtained by subtracting $\kappa_e$ and $\kappa_B$ from $\kappa_T$ based on the formula $\kappa_L = \kappa_T - \kappa_e - \kappa_B$ (Fig. 5b), in which the charge carrier thermal conductivity $\kappa_e$ is estimated according to the Wiedemann-Franz law ($\kappa_e = LT\sigma$) (Fig. S9). The Lorenz number ($L$) was calculated by deriving the Fermi energy from the Seebeck coefficient with an assumption of a single parabolic band model and acoustic phonon scattering, as illustrated in Fig. S10[22,66]. Additionally, bipolar thermal conductivity ($\kappa_B$) should be considered for narrow bandgap semiconductors. To clarify the contribution of $\kappa_B$, we separated the $\kappa_B$ from the $\kappa_T$ according to the method previously reported[67]. The relationship between $\kappa_T - \kappa_e$ and $T^{-1}$ is given in Fig. S11. Usually, $\kappa_L$ is estimated by the expression $\kappa_L = 3.5(\frac{k_B}{h})MV^{4/3}\theta_D^3\gamma^{-2}T^{-1}$, where $k_B$, $h$, $M$, $V$, $\theta_D$ and $\gamma$ are the Boltzmann constant, Planck constant, average mass per atom, average atomic volume, Debye temperature, and Grüneisen parameter, respectively[68]. According to this relation, $\kappa_L$ is linearly dependent on $T^{-1}$. As a consequence of the contribution of bipolar thermal diffusion, $\kappa_L$ is always overestimated in the intrinsic conduction region[67]. At high temperatures, the slight deviation from the linear relationship of $\kappa_T - \kappa_e$ for $Sn_{0.8}Al_{0.08}Sb_{0.15}Te$ and $Sn_{0.8}Al_{0.08}Sb_{0.15}Te-4\%AgBiTe_2$ samples can be attributed to the bipolar effect. This is ascribed to the reduced band gap (Table S3) and carrier concentration. Results show that $\kappa_L$ of Al and Sb codoped samples is reduced compared with the pristine SnTe and $Sn_{1.03}Te$ samples. In particular, $\kappa_L$ of all $AgBiTe_2$-containing samples is further reduced compared to that of $Sn_{0.8}Al_{0.08}Sb_{015}Te$. Ultimately, an ultralow $\kappa_L$ of ~ 0.32 $Wm^{-1}K^{-1}$ is achieved at 873 K for $Sn_{0.8}Al_{0.08}Sb_{015}Te-4\%AgBiTe_2$, which is not only below the theoretical amorphous limit (~0.4 $W\,m^{-1}\,K^{-1}$) of SnTe calculated from the Debye−Callaway model[42] but also lower than most of the reported values for SnTe-based thermoelectrics[22,34,55,69–73] (Fig. 5c). However, this

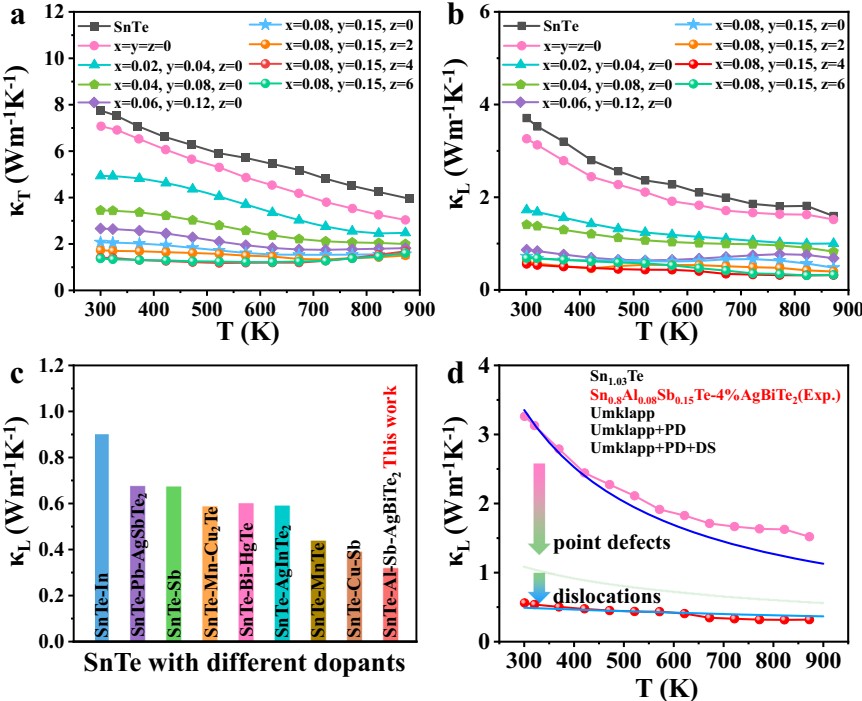

**Fig. 5 | Thermal transport properties of $Sn_{1.03-x-y}Al_xSb_yTe-z\%AgBiTe_2$ samples.** Temperature dependence of **a** total thermal conductivity $\kappa_T$; **b** lattice thermal conductivity $\kappa_L$ of $Sn_{1.03-x-y}Al_xSb_yTe-z\%AgBiTe_2$ samples. **c** Comparison of $\kappa_L$ of $Sn_{0.8}Al_{0.08}Sb_{0.15}Te-4\%AgBiTe_2$ with other reported SnTe systems[22,34,55,69–73]. Estimation of dislocation density in $Sn_{0.8}Al_{0.08}Sb_{0.15}Te-4\%AgBiTe_2$ by MWH method, **d** Temperature-dependent lattice thermal conductivity fitted by the Callaway model with different phonon scattering sources; the experimental $\kappa_L$ values of $Sn_{1.03}Te$ and $Sn_{0.8}Al_{0.08}Sb_{015}Te-4\%AgBiTe_2$ are provided for comparison.

value is higher than the amorphous limit of SnTe predicted incorporating the Born-von Karman periodic boundary conditions (-0.12 W m$^{-1}$ K$^{-1}$)[74].

**Microstructure observations in $Sn_{0.8}Al_{0.08}Sb_{0.15}Te-4\%AgBiTe_2$**

To further understand the mechanism of this ultralow $\kappa_L$, microstructural characterizations of the $Sn_{0.8}Al_{0.08}Sb_{0.15}Te-4\%AgBiTe_2$ were investigated with scanning transmission electron microscopy (STEM) in both annular bright field (ABF) and high-angle annular dark-field (HAADF) imaging modes. Low-magnification ABF-STEM images reveal high-density dislocation networks (Fig. 6a). Medium-magnification ABF-STEM images (Fig. 6b) demonstrate that these dislocations tend to intertwine into dislocation networks. Figure 6c and d show the corresponding HAADF-STEM image and EDS elemental mappings from the yellow-frame area in Fig. 6b, highlighting the homogeneous chemical compositions around dislocations. Typical edge dislocations can be identified, and the corresponding projected Burgers vectors are 1/2[010] and 1/2[00_1], as confirmed by the Burgers loop (Fig. 6e–g). The geometric phase analysis (GPA), which is a semiquantitative lattice image processing approach, is applied to map the spatially distributed strain field caused by the dislocations (shown in Fig. 6f–h). Strong average lattice strain fluctuations are induced by the dense dislocations, which can significantly shorten the phonon relaxation time and reduce the lattice thermal conductivity[75,76]. These high-density dislocations could be nucleated and propagated due to the strong lattice distortions induced by the large variation of atomic size of constituent elements[77]. In this work, dopant atoms such as Al$^{3+}$ (0.54 Å) and Sb$^{3+}$ (0.76 Å) are much smaller than Sn$^{2+}$ (0.93 Å), especially Al, with a volume of only 1/3 that of Sn. The resulting stress field serves as a dynamic driving force for the formation of dislocations. Furthermore, the arrangement of doping atoms plays a pivotal role in both lattice distortion and dislocation formation. As illustrated in Fig. 3c, atoms doped at Sn positions (Al, Sb, Ag, and Bi) exhibit a discernible level of

disorder, particularly evident in the case of the Al atom due to its significantly smaller volume compared to Sn. This intricate interplay between short-range order and long-range disorder in atomic arrangement facilely triggers the formation of dislocations[78,79]. Beyond dislocation nucleation, doping also exerts influence on the diffusion and migration of dislocations. The diffusion of doping atoms promotes the generation of additional dislocations, thereby contributing to the increased dislocation density. The synergistic effects of all these factors collectively account for the observed high-density dislocations.

The modified Williamson-Hall (MWH) method is used to estimate the dislocation density of $Sn_{0.8}Al_{0.08}Sb_{015}Te-4\%AgBiTe_2$. Details of the calculation can be found in Supporting Information. The dislocation density in $Sn_{0.8}Al_{0.08}Sb_{0.15}Te-4\%AgBiTe_2$ can reach up to $2.9 \times 10^{11}$ cm$^{-2}$. This value is consistent with the geometrical necessary dislocation (GND) density ($1 \times 10^{11}$ cm$^{-2}$, Fig. S12) measured using electron backscatter diffraction (EBSD). Note that this small difference in the dislocation density is reasonable and acceptable due to the different characterization techniques. Moreover, the above STEM images give a dislocation density of -8.7 $\times 10^{10}$ cm$^{-2}$, approaching the order of $10^{11}$ cm$^{-2}$, slightly lower than the XRD and EBSD results. This discrepancy is attributed to the influence of factors such as uneven dislocation distribution, dislocation extinction, and sample surface mirroring forces during the STEM characterization process, which typically lead to an underestimation of dislocation density values. Overall, the three sets of values collectively confirmed that the dislocation density of the synthesized material is approximately in the order of $10^{11}$ cm$^{-2}$, surpassing by more than one order of magnitude the values reported for SnTe-based materials in the past, such as a dislocation density of $3.46 \times 10^{10}$ cm$^{-2}$ in $Sn_{0.995}In_{0.005}Te$[80]. According to the phonon scattering model developed by Klemens[75,81], we have calculated the temperature-dependent lattice thermal conductivity of SnTe with phonon scattering sources of Umklapp processes, point defects, and dislocations, as illustrated in Fig. 5d. Details of the

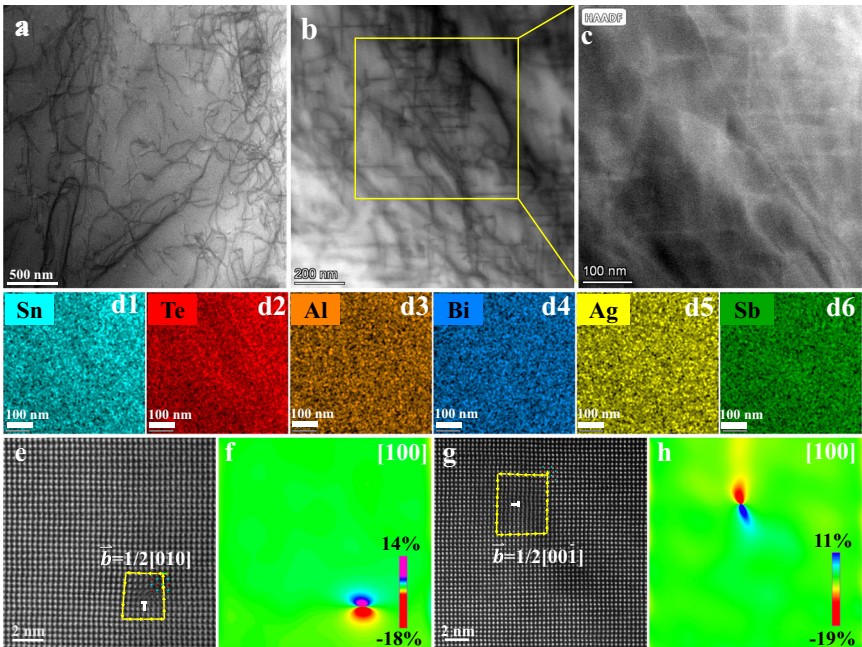

**Fig. 6 | Microstructural characterizations of $Sn_{0.8}Al_{0.08}Sb_{015}Te$-4%AgBiTe$_2$ samples. a, b** ABF-STEM images showing the dislocations (the darker lines), **c** is the close-up of the HAADF-STEM image of **b**. **d1-d6** EDS mappings taken from **c**. **e, g** Atomic-resolution HAADF-STEM image of the dislocation core, and **f, h** Strain mapping corresponding to the STEM image of **e, g** using the geometrical phase analysis.

calculation can be found in Supporting Information. Due to the large atomic size and mass difference between Al, Sb, and Sn, the lattice thermal conductivity of SnTe is significantly decreased upon the formation of Al and Sb point defects. The addition of Ag and Bi point defects in SnTe ($Sn_{0.8}Al_{0.08}Sb_{0.15}Te$-4%AgBiTe$_2$) can only slightly decrease the lattice thermal conductivity due to the small concentration (4%). Once taking the dense dislocations into account ($Sn_{0.8}Al_{0.08}Sb_{0.15}Te$-4%AgBiTe$_2$+dislocation), the local strain field can markedly reduce the lattice thermal conductivity over the entire temperature range studied, in good agreement with the experimental data. This confirms that dense dislocations play a significant role in minimizing the lattice thermal conductivity[24,82]. To clarify the frequency dependence of phonon scattering by different scattering sources, the spectral lattice thermal conductivity at room temperature is calculated and illustrated in Fig. S13. The results demonstrate that point defects scatter high-frequency phonons while dislocations mainly scatter mid-frequency phonons.

## ZT values and energy conversion efficiency of the thermoelectric device

The significantly enhanced PF in conjunction with the suppressed lattice thermal conductivity leads to remarkably high thermoelectric performance over a broad temperature range, as shown in Fig. 7a. The average ZT ($ZT_{ave}$) of 1.15 is achieved for $Sn_{0.8}Al_{0.08}Sb_{0.15}Te$-4%AgBiTe$_2$ in the range from 300 K to 873 K (Fig. 7b), which is a record-high value for SnTe systems[10,22,42,55,69,71,83–86]. Thermoelectrics find wide applications in this temperature range such as for industrial waste heat recovery, high-temperature sensors power supply, and deep-space missions. This value is also superior to many other Pb-free thermoelectric compounds, as summarized in Table S4. The maximum ZT value of $Sn_{0.8}Al_{0.08}Sb_{0.15}Te$-4%AgBiTe$_2$ is competitive among previously reported SnTe-based materials (Fig. S14). Notably, the near room temperature ZT value reaches 0.36 at 300 K, showing a pronounced enhancement compared to other SnTe thermoelectrics (Figs. S15–S17). Multiple measurements confirm that this performance is very stable and reproducible (Figs. S18–S19). The significant

enhancement of properties over a broad temperature range is mainly ascribed to the optimized carrier concentration by increasing the m*, which can be visualized in the figure showing the reduced Fermi level ($\eta = E_F/k_BT$) dependent ZT by changing the B factor[24] (Fig. 7c, d). The $\eta$ value is approaching the optimum value at both 300 and 673 K by introducing extra DOS hump and band convergence that increase the m*. We still notice a big space for further ZT enhancement, as illustrated in Fig. 7c, d, by either reducing the carrier concentration or increasing the m*. The large $ZT_{ave}$ yields a high theoretical conversion efficiency of 16.7%, which is simulated for the single-leg $Sn_{0.8}Al_{0.08}Sb_{0.15}Te$-4%AgBiTe$_2$ module at a temperature difference ($\Delta T$) of 573 K (Fig. S20). This theoretical value outperforms most of the high-performance thermoelectric systems. A prototype thermoelectric device with 17 couples was assembled using our $Sn_{0.8}Al_{0.08}Sb_{0.15}Te$-4%AgBiTe$_2$ as p-type legs and commercially available Bi$_2$Te$_3$ as n-type legs. We choose commercial Bi$_2$Te$_3$ as the n-type legs because it has a similar compatibility factor to SnTe since the materials combined in a high-performance thermoelectric generator (TEG) need similar compatibility factors[87]. Experimentally, we achieved a high output power of 662 mW and a high energy conversion efficiency of 5.4% under a temperature difference $\Delta T$ of 350 K (Fig. 7e). This value is 80% larger than that of the commercial (Bi, Sb)Te$_3$ (Fig. 7f) and even higher than other typical thermoelectric devices including SnTe, SnSe, PbTe and PbTe-based LAST crystal at the same temperature range, as demonstrated in Fig. 7f[88–94]. We also noticed that this experimental conversion efficiency is much lower than that simulated for the single-leg $Sn_{0.8}Al_{0.08}Sb_{0.15}Te$-4%AgBiTe$_2$. This is because the theoretical value in Figure S20e is simulated under an ideal condition without the effects of n-type material and interfacial properties. Yet, the interfacial properties were proven to have around a 20% influence on the conversion efficiency of a TEG[95]. Also, during the efficiency measurement, thermal radiation significantly impacts the measured heat flow Qc, which is difficult to measure accurately. As a result, the heat flow through the device was overestimated and finally led to a lower actual efficiency[96,97]. This implies that further improvement of the device efficiency can be achieved by better designing the

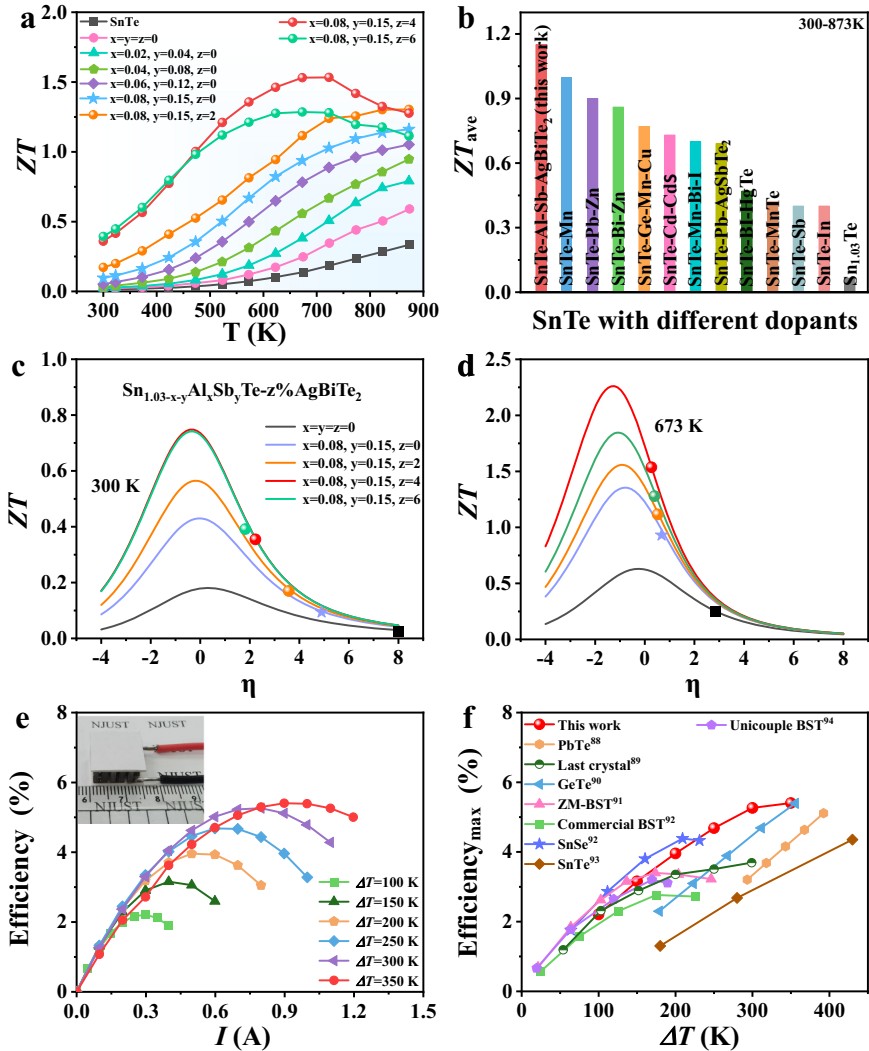

**Fig. 7 | Thermoelectric figure-of-merit and energy conversion performance.**
**a** $ZT$ of $Sn_{1.03-x-y}Al_xSb_yTe-z\%AgBiTe_2$ as a function of temperature. **b** Comparison of $ZT_{ave}$ of $Sn_{0.8}Al_{0.08}Sb_{015}Te-4\%AgBiTe_2$ with other reported SnTe systems[10,22,42,55,69,71,83–86]. Maximum $ZT$ values as a function of the Fermi level ($\eta$). The corresponding compositions are given in the figures represented by various colors matching the data curve: **c** at 300 K, **d** at 673 K. **e** Measured energy conversion efficiency as a function of current $I$ of the 17 pair thermoelectric device. The slightly lower efficiency under $\Delta T = 350$ K and low currents could be due to the poor thermoelectric performance of n-type $Bi_2Te_3$ in this temperature range; **f** Comparison of measured thermoelectric conversion efficiencies for devices in this study (fabricated using our $Sn_{0.8}Al_{0.08}Sb_{0.15}Te-4\%AgBiTe_2$ as p-type legs and commercially available $Bi_2Te_3$ as n-type legs) and PbTe[88], PbTe-based LAST crystal[89], GeTe[90], (Bi, Sb)Te_3 (BST)[91,92,94], SnSe[92], SnTe[93] systems among comparable temperature differences $\Delta T$ (ZM, zone melting).

interfacial properties of the device[98] and increasing the thermoelectric performance of the n-type counterparts. Nevertheless, we realized a high efficiency under such a small temperature difference, highlighting the important application of SnTe-based TE materials in the field of low-grade heat recovery. Excellent mechanical properties are indispensable for well-established thermoelectric materials. Microhardness and nanoindentation measurements were performed on $Sn_{1.03}Te$ and $Sn_{0.8}Al_{0.08}Sb_{0.15}Te-4\%AgBiTe_2$. Vickers hardness and hardness (H) of $Sn_{0.8}Al_{0.08}Sb_{0.15}Te-4\%AgBiTe_2$ are both improved over pristine $Sn_{1.03}Te$, as shown in Fig. S21. The Vickers hardness increases from 65.6 HV0.1 for $Sn_{1.03}Te$ to 170.7 HV0.1 for $Sn_{0.8}Al_{0.08}Sb_{0.15}Te-4\%AgBiTe_2$. Meanwhile, a high hardness value of 2.5 GPa is achieved in $Sn_{0.8}Al_{0.08}Sb_{0.15}Te-4\%AgBiTe_2$, which is 132% higher than pristine $Sn_{1.03}Te$. This significantly enhanced mechanical properties can be attributed to the high density of dislocations observed. The enhancement of the mechanical properties of $Sn_{0.8}Al_{0.08}Sb_{0.15}Te-4\%AgBiTe_2$ enhances the stability of the thermoelectric properties during preparation and application and is also beneficial for thermoelectric devices to resist external mechanical shocks.

## Discussion

This study demonstrates a feasible way to replace PbTe with SnTe upon considerably increasing the average $ZT$ value of SnTe. This goal has been realized by doping Al and alloying $AgBiTe_2$ with SnTe to manipulate the electronic band structures. We demonstrate that the contribution of Sn-s electronic states at the Fermi level stems from the intrinsic metavalent bonding mechanism of SnTe, which leads to special symmetry and orbital phases at the L point that create occupied anti-bonding states just below the Fermi level. Following the design rule of spatial and energetic overlap between the s-orbital of dopants and the p-orbital of Te, we have screened out Al as an efficient candidate to create an extra DOS hump at the Fermi level. This band modification strategy in conjunction with alloying $AgBiTe_2$ to increase the valley degeneracy improves the density-of-states effective mass and, thus, enlarges the optimum carrier concentration required at low temperatures, which can be more easily realized in SnTe with an intrinsically high carrier concentration. The optimization of carrier concentration leads to a significant enhancement of power factor in the whole temperature range. The $Sn_{0.8}Al_{0.08}Sb_{0.15}Te-4\%AgBiTe_2$

sample exhibits a maximum PF of 28.1 $\mu$W cm$^{-1}$ K$^{-2}$ at 573 K. High-density dislocations are introduced in SnTe through strong lattice distortions induced by the large mismatch in the size of doping elements. This significantly reduces the lattice thermal conductivity. An ultralow $\kappa_L$ of ~ 0.32 Wm$^{-1}$K$^{-1}$ is achieved at 873 K for Sn$_{0.8}$Al$_{0.08}$Sb$_{015}$Te–4%AgBiTe$_2$, which is lower than most of the SnTe-based thermoelectrics reported. The reduced lattice thermal conductivity and improved power factor finally contribute to a high $ZT$ of 0.36 at 300 K and an average $ZT$ of 1.15 between 300 and 873 K. Obtaining a large average $ZT$ over a broad temperature interval is of more significance than just achieving a high peak $ZT$ value since the final energy conversion efficiency depends on the average performance. We also achieved a high efficiency of 5.4% in a 17-couple prototype thermoelectric module at a temperature difference of 350 K, highlighting the important application of SnTe-based TE materials in low-grade heat recovery. Our study provides a new approach to optimize the thermoelectric performance at low to intermediate temperatures for materials with intrinsically high carrier concentrations.

## Methods

### Synthesis

High-purity single elements Sn (99.99%, powder), Te (99.99%, powder), Al (99.99%, granular), Sb (99.99%, powder), Bi (99.99%, powder) and Ag (99.99%, powder) were sufficiently mixed by manual-grinding according to the calculated compositions of Sn$_{1.03-x-y}$Al$_x$Sb$_y$Te (x = 0, 0.02, 0.04, 0.06, 0.08 and y = 0, 0.04, 0.08, 0.012, 0.015, 0.18), and Sn$_{1.03-x-y}$Al$_x$Sb$_y$Te-z%AgBiTe$_2$ (molar fraction in the text, z = 2, 4, 6). The evenly mixed powders were then loaded inside high-temperature resistant quartz tubes with an inner diameter of 17 mm. The tubes were sealed in a vacuum of about 10$^{-5}$ mbar and then put into a high-temperature furnace heated to 1273 K in 10 h. The melt was kept at this temperature for 8 h for homogenization. After that, the melt was quenched into room-temperature water and thus an ingot was obtained. We then use mortar and pestle to grind the ingot into powders with uniform size distributions. These powders were loaded into a graphite die and then sintered by a spark plasma sintering system (HPD 10, FCT System GmbH) at 873 K for 7 min under an axial pressure of 50 MPa in a vacuum of about 12-18 Pa.

### Characterization

The powder X-ray diffraction patterns were determined by using a Bruker D8 Advance instrument with Cu K$\alpha$ radiation ($\lambda$ = 0.154060); the scanning step size was set as 0.02 and the average scanning time was 0.1 s. An Ultra-High Resolution Scanning Electron Microscope Gemini SEM 500 was performed to investigate the structural morphology, while the attached Aztec Extreme Energy Dispersive Spectrometer (EDS) was used for elemental mapping. High-angle annular dark field scanning transmission electron microscopy (HAADF-STEM) images were collected using a FEI Titan Themis Z transmission electron microscope (TEM) equipped with a double corrector. Specimens for high-resolution STEM observations were prepared through traditional mechanical polishing, dimpling, and argon ion milling with a liquid nitrogen stage. Atom probe tomography (APT) needle-shaped specimens were fabricated using a dual-beam SEM/focused ion beam (Helios NanoLab 650, FEI) following the standard "lift-out" method. Measurements were conducted on a local electrode atom probe (LEAP 5000XS, CAMECA) utilizing 10-ps, 10-pJ ultraviolet laser pulses (wavelength = 355 nm) with an average detection rate of 1%, a pulse repetition rate of 200 kHz, and a base temperature of 40 K. The ion flight path was set at 100 mm and the detection efficiency was limited to 80%. Data reconstruction and analysis were performed using AP Suite 6.1.

### Transport properties measurement

The sintered ingots were cut and polished into rectangular blocks of 8 × 3 × 3 mm$^3$ to measure the Seebeck coefficient ($S$) and electrical conductivity ($\sigma$) on an Ulvac-Riko ZEM-3 instrument system from 300 K to 873 K. The measurements were conducted under a helium atmosphere and the uncertainty is about 5%. The total thermal conductivity ($\kappa_T$) was determined by using the equation of $\kappa_T = DC_p\rho$. We use a laser flash instrument (Netzsch LFA-457) to receive the thermal diffusivity ($D$) and the uncertainty is about 5%. By using the density meter (ME204E), we evaluated the density of the samples ($\rho$) by Archimedes method with an uncertainty of ~2%, which is shown in Table S5. The degree of lattice distortion becomes increasingly severe with increasing doping content contributing to the reduction in the relative density of the synthesized products as the doping content increases. The specific heat capacity ($C_p$) was indirectly derived from a reference in the range of 300–873 K[99]. The Hall coefficient ($R_H$) was obtained from the Hall measurement instrument (HMS-3000) using the van der Pauw method at 300 K. Through the two formulas $n = 1/(eR_H)$ and $\mu = \sigma R_H$, we can acquire the room-temperature hall carrier concentration ($n$) and carrier mobility ($\mu$). In general, the combined accuracy for the calculation of $ZT$ is about 20%.

### DFT calculations

Density functional theoretical (DFT) electronic structure calculations are carried out using Vienna ab initio simulation package (VASP)[100–102] code based on the plane-wave projector augmented wave (PAW) scheme. A Generalized Gradient Approximation (GGA) to exchange-correlation energy with the functional of Perdew, Burke, and Erzenhoff (PBE) is used to perform the DFT calculations[103]. An energy cut-off of 450 eV is applied for the plane-wave expansion in both primitive cell and supercell calculations, ensuring that the total energy converges within 10$^{-5}$ eV. The Brillouin zone of the SnTe primitive cell is sampled using a (15 × 15 × 15) Monkhorst−Pack k-point grid[104]. All structures are fully optimized until the maximum force on each atom is below 0.01 eV. To examine the effects of doping on the band structure and chemical bonding of SnTe, a large supercell of SnTe is constructed (dimensions: a = 13.61 Å, b = 13.61 Å, c = 13.61 Å, containing 27 Sn and 27 Te atoms) to simulate the experimentally optimized doping concentrations. Various initial configurations of Al and (Bi, Ag) dopants in the SnTe matrix are explored to identify the most stable configurations, characterized by the lowest DFT energy. Replacing one Sn atom (1 dopant and 26 Sn) with the dopant atom based on this supercell is used to calculate the density-of-states of the doped system. The Brillouin zone of the supercells is sampled using a (3 × 3 × 3) Monkhorst−Pack $k$-point grid. For calculating the band structures of a supercell derived from the corresponding primitive cell, we apply a band folding method. To further examine the band structures of the supercell, we employ a band unfolding technique using the BandUP code[105], which recovers the effective primitive cell band structure along the high-symmetry directions of the primitive cell. Chemical bond analysis including crystal orbital bond index (COBI) is performed using the LOBSTER program[106].

### Thermoelectric device fabrication and measurement

The thermoelectric (TE) device contains 17 couples of p-n legs with an overall size of 15 mm × 15 mm × 5 mm. The p-type legs are the preparative Sn$_{0.8}$Al$_{0.08}$Sb$_{015}$Te–4%AgBiTe$_2$ samples while the n-type counterparts are the commercial Bi$_2$Te$_3$ samples prepared by zone-melting. The TE transport properties of the n-type legs are summarized in Table S6 of the Supporting Information. Both p-type and n-type TE legs were coated with an electroplated Ni layer as a diffusion barrier. The legs were then cut to a cross-sectional size of 1.5 × 1.5 mm$^2$ and a length of 5.0 mm before being soldered to two direct-bonded copper alumina ceramics using Sn$_5$Pb$_{92.5}$Ag$_{2.5}$ solder. The conversion efficiency of the TE module was measured using a custom-built testing system.

## Mechanical properties

The hardness measurements were performed on polished sample surfaces with a microhardness tester (Johoyd HVS-1000Z) and a nanoindenter (G200).

## Reporting summary

Further information on research design is available in the Nature Portfolio Reporting Summary linked to this article.

## Data availability

All data necessary to understand and assess this manuscript are shown in the main text and the Supporting Information. The data that support the findings of this study are available from the corresponding author upon reasonable request.

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

## Acknowledgements

The work was supported by the National Natural Science Foundation of China (Nos. 52071182 and 52202049). Y. Zhang acknowledges financial support from the program "Distinguished Expert of Taishan Scholar" (No. tstp20221124). X. Zhang acknowledges the Natural Science Foundation of Ningxia Province (Grant No.2021AAC03241). The authors acknowledge the computational resources granted from RWTH Aachen University under project p0021179.

## Author contributions

G.D.T., Y.Y., and M.W. conceived and designed the experiments. G.D.T., Y.Q.L., and Y.Y. wrote the draft. Y.Q.L. prepared samples and analyzed data. X.M.Z. Y.S.Z. and P.Y. carried out the DFT calculations for band structures. Y.X.Y., P.F.N., and B.H.G. accomplished the microstructural characterizations. Y.R.G., X.F.M., and K.S. helped measure the thermoelectric properties. N.L. prepared the APT specimens and carried out the APT characterization. Y.Y. analyzed the APT data. D.K. carried out DFT calculations for the chemical bonding analyses. C.F.S. helped Y.Y to prepare Figs. 1 and 2. M.C. provided helpful discussions on the phase of orbitals. Y.Y. and M.W. established the relationship between electronic band structures and chemical bonds. All authors have discussed the results and approved the submission.

## Funding

## Competing interests

The authors declare no competing interests.
