## [Transparent Peer Review file · Nature Communications]

Interplay between metavalent bonds and dopant orbitals enables the design of SnTe thermoelectrics

Corresponding Author: Dr Yuan Yu

Version 0:

Reviewer comments:

Reviewer #1

(Remarks to the Author)

Please find my reviews in the attached .pdf file.

Reviewer #2

(Remarks to the Author)

In this work the thermoelectric performance of SnTe has been improved by co-doping Al and Sb and further alloying AgBiTe₂. The work focuses on the electronic structure tuning with an aim to increase the DOS effective mass. The work is good but needs to be revised adding additional data. Below are some of the points to be considered.

The introduction should give a brief history of various dopants used to improve the DOS effective mass like V, W, Mo. Not only indium but there are other dopants like Bi, Zn which have the potential to introduce resonant states in SnTe. These dopants have versatile effects leading to increase in band gap, hyperconvergence, Rashba effect, introduction of multiple electronic valleys etc not only in SnTe but also in other telluride systems. Details on these need to be mentioned as electronic structure engineering is the approach adopted in this paper.

The authors write "We have previously demonstrated that Sb doping produces negligible changes to the electronic structure of SnTe [41], which has also been shown and interpreted in Figure 2." This is not accurate as doping of Sb has shown to introduce additional energy levels in the band gap region in SnTe. The authors need to modify the statement. Electronic structure of Sb doped SnTe should be provided. The authors mention the term "We" which previous work are the authors referring to?

Electronic structures for Al doped SnTe, Al and Sb co-doped SnTe and Al and Sb co-doped SnTe with AgBiTe₂ alloyed composition should also be provided.

Some of the samples show onset of bipolar conduction. Electronic structure should be used to explain this.

Ultra low lattice thermal conductivity has been observed in Cu and Sb doped SnTe also which is slightly higher (0.39 W/mK) than the present value. This should be included in the bar diagram.

The authors write "The average ZT (ZT_{ave}) of 1.28 is achieved for Sn_{0.8}Al_{0.08}Sb_{0.15}Te–4%AgBiTe₂ in the range from 400 K to 873 K (Figure 7b), which is a record-high value for SnTe systems^{22, 41, 55-58, 71}." There are reports which show higher average zT values than the ones cited here. The authors need to do a thorough literature survey and compare values with the highest values reported by far.

Why is the value of zT at 373 K highlighted in the abstract and conclusion? It is better to indicate the value at 300 K which is the lowest temperature of the study. Similarly for calculating average zT the authors should consider 300 K instead of 400 K. It is better to update the references to include the ones published in the last 5 years to indicate the recent developments.

Reviewer #3

(Remarks to the Author)

The authors report the enhancement of zT in SnTe by doping Al and Sb and alloying it with AgBiTe₂.

The rationale of the work is good but it can be improved further by considering the points below.

The introduction concentrates more on the carrier tuning than the electronic structure part. To give a clear state of art in the field of SnTe thermoelectrics it is essential to briefly mention various approaches adopted to improve the zT by tuning the electronic structure like perfect convergence of light and heavy hole valence bands and light and heavy electron conduction bands, hyperconvergence of valence bands and conduction bands, Rashba splitting, band gap increase etc. attained by doping.

Various dopants and their impact on the electronic structure has to be provided in a tabular column.

The authors need to clearly indicate what is the reason behind choosing AgBiTe₂ for alloying. As doping of Ag would have led to band convergence.

The electronic structure should be provided for the configuration showing maximum activity. That is SnTe doped with Al and Sb and Sn vacancies occupied by Ag and Bi.

Alloyed samples show onset of bipolar conduction in Figure 4c. Co-relation must be done with the electronic structure results.

The authors need to clearly indicate what is the amount of Ag and Bi which enter into the lattice site of the host and what is the amount which remains undoped.

A comparison plot (bar diagram) of room temperature (300 K) zT values should be provided with previous reports of high value.

A comparison plot (bar diagram) of peak zT values with previously reported high performing SnTe materials (zT above 1.5) should be given.

Average zT should be reported between 300 K and 873 K.

Have the authors tested the mechanical properties of the sample?

The authors need to take care of grammatical and spelling errors in the manuscript and supplementary information during revision.

Version 1:

Reviewer comments:

Reviewer #1

(Remarks to the Author)

I found that the authors have adequately revised the manuscript addressing my comments. I recommend the article to be accepted for the publication.

Reviewer #2

(Remarks to the Author)

The authors have thoroughly revised the manuscript. The manuscript in present form can be accepted for publication.

Reviewer #3

(Remarks to the Author)

The authors have now revised the manuscript as suggested and the same may be accepted.

Reviewer #1:

The manuscript entitled ‘Interplay between metavalent bonds and dopant orbitals enables the design of SnTe thermoelectrics’ presents a way of manipulating electronic structures to optimize the thermoelectric efficiency by doping Al and alloying AgBiTe₂ with SnTe. Overall, this work, with combined experimental and theoretical approach, looks extensive, well-organized, most of the relevant physical properties/parameters calculated, and results are promising for the TE device applications. So, I suggest the article to be published in this journal. However, I have following comments to be addressed in the revised manuscript before the acceptance and publication in this journal:

Response:

We appreciate your positive and encouraging comments. We have provided a point-by-point response below and have revised the manuscript according to your suggestions.

Comment: 1. Figure 1c seems to be the key point to identify/prove metavalent bonding in SnTe. But it is not clear how the authors come up with the scheme in Figure 1c, i.e. how did SnTe show half-filled p-orbitals? I suggest adding more details/analyses on it.

Response:

You are perfectly right with your conclusion, half-filled p-orbitals are the key to unraveling the bonding and the resulting electronic structure of SnTe. The scheme in Fig. 1c is a combination of the electronic band structures (physical perspective) and the phase of orbitals at different points (chemical perspective) in the first Brillouin zone. The shape of the electronic band structures is obtained by DFT calculations as shown in Fig. 1b. There the wave function corresponding to each band is projected onto the relevant orbitals. In the vicinity of the Fermi levels, these orbitals are the Sn 5s and 5p states, as well as the Te 5s and 5p states. The corresponding contributions are depicted in the orbital resolved density of states (DOS) in Fig. 1a. This figure shows that the Sn 5s-state is filled and hence can not contribute to bonding, while the Te 5s-state is almost completely filled and hence also not relevant to bonding (Fig. 1d). The bonding is hence governed by the Sn p-orbitals and the Te p-orbitals (Fig. 1d). The phase of different orbitals is determined by the wavefunctions of s- and p-orbitals at different k -points (Γ and L). These k -points are particularly relevant to understand bonding and properties in SnTe. According to the Bloch theorem, a certain atomic orbital is in-phase in adjacent atoms of the same type at the Brillouin zone center (Γ point). For example, the phase of Te-p or Sn-p orbitals repeats in neighboring unit cells as sketched at the Γ point in Fig. 1c. This leads to the bonding configuration if the same phase orbitals are head-to-

head between Te-p and Sn-p and otherwise the anti-bonding configuration. Different from the wavefunction of p-orbital which has a shape depicted as lobes and has a nodal plane, the wavefunction of s-orbital is spherical. This gives rise to the canceling effect between the bonding and anti-bonding states between the s-orbital of one atom (Sn or Te) and its adjacent p-orbitals of the other (Te or Sn). In contrast, at the L point, a certain atomic orbital, e.g., Te-p, is out-of-phase in neighboring unit cells. This leads to pure anti-bonding configurations for the Sn-s and Te-p orbitals and allows the contribution of Sn-s to the electronic structures, as illustrated at the L point in Fig. 1c. These are the main ideas and key information shown in Fig. 1c.

Regarding the half-filled sigma-bond formed by p-orbitals, we indeed did not provide detailed explanations. As mentioned above, the results shown in Fig. 1d (obtained by COBI calculations) indicate that the Sn-s orbitals do not contribute to the bonding formation because the fully occupied anti-bonding state and the bonding state cancel each other, as can also be reflected by the negligible ES value between Sn-s and Te-p (0.01). Thus, the bonding is dominated by the p-states. Since SnTe has in total 6 p-electrons and an effective coordination number of 6, the average number of p-electrons for each bond is just one. This is different from the electron pair required for a typical covalent bond described by Lewis. Accordingly, SnTe shows half-filled p-states and this special bonding mechanism is called metavalent bonding.

Revisions: We have described Fig. 1c in more detail in the revised manuscript on page 8. The reason for the formation of half-filled p-states in SnTe has also been added and highlighted on page 10.

Comment: 2. About Figure 2, I suggest authors to add more detailed physical insights behind the usefulness of so called ‘golden rectangle’. And what is/are the significance(s) of symbol size vs size of s-orbital of the dopants? Crucially, why the symbol size is scaled to difference between 0 eV and -0.1 eV, NOT -0.05 eV, -0.15 eV, ...? It could change things I believe.

Response:

The conclusions and physical insights of Fig. 2 are based on the findings presented in Fig. 1. Since the Fermi energy level in SnTe is located near the L point of the valence band, the electrical transport properties can be tailored by tuning the electronic states at the L point. As we have explained in Fig. 1 by combining both physical and chemical

perspectives, the Sn-s orbital contributes to the electronic states at the L point by hybridizing with the Te-p orbital. Thus, the density-of-states at the L point can be modified by changing the degree of orbital overlap between cation-s and Te-p through substituting Sn with various dopants. This degree of overlap depends on both the spatial size and energy differences between dopant-s and Te-p orbitals. This is why we use the “size of s-orbital” and the “energy level of s-states” as the x and y axis in Fig. 2. It is expected that the smaller difference in the size and energy between dopant-s and Te-p states, the larger the orbital overlap between these states, and the stronger the DOS contribution (‘hump’) at the Fermi level. The “golden rectangle” in Fig. 2 is built based on these understandings to ensure the maximum orbital overlap.

The symbol size is scaled to the integrated partial DOS (iDOS) of SnTe with corresponding dopants between 0 eV and -0.1 eV. Here, the Fermi energy level is defined at 0 eV in the DFT calculations. Thus, we start from 0 eV, not -0.05 eV. The cutoff energy of -0.1 eV is chosen based on the fact that only the states near a few $k_B T$ (one $k_B T$ is about 25 meV at room temperature) of the Fermi energy can contribute to the charge transport. As we can see in Fig. S2 (also copied here as Fig. R1 for your convenience), a small change in the energy range starting from the Fermi energy level to a few $k_B T$ will not change the value of iDOS significantly. This supports the robustness of the relative trend of the symbol size in Fig. 2. In principle, the symbol size should increase with decreasing the difference in size between the dopant-s and Sn-s orbitals. This partly explains the much larger symbol size for Al and In, which are closer to the “golden rectangle”. Yet, besides the orbital size, the energy of the dopant-s orbital is also very important. Again, those dopants located between Sn-s and Te-p in the y-axis show larger symbol sizes. Overall, the final symbol size is determined by the synergistic effect of the size and energy of the dopants, which can be nicely described in Fig. 2.

Fig. R1. Integrated partial density-of-states (iDOS) for SnTe with various dopants. The symbol size in Fig. 2 of the main manuscript scales to the difference between the iDOS at 0 eV and -0.1 eV.

Revisions: We have added a more detailed explanation of the insights to obtain the “golden rectangle” from the perspective of maximizing the overlap between dopant-s and Te-p overlaps by considering both the spatial and energetic overlaps. We have also explained why the energy range between 0 eV and -0.1 eV was chosen to calculate the difference in iDOS on page 11.

Comment: 3. In Figure S4a, y% refers to Sb element. However, in Figure S4b the plot shows y%, but the caption has Te-z% without y in the provided composition/compound. It is not clear.

Response:

We appreciate the comment from the reviewer. Fig. S4b is the lattice parameter as a function of AgBiTe₂ fraction in Sn_{0.8}Al_{0.08}Sb_{0.15}Te-z%AgBiTe₂ (z=2, 4, 6). We have changed the x-axis label to z% in Fig. S4b (Fig. R2).

Revisions: We have revised Fig. S4b in the Supporting Information.

Fig. R2. Lattice parameter as a function of AgBiTe₂ fraction in Sn_{0.8}Al_{0.08}Sb_{0.15}Te–z%AgBiTe₂ (z=2, 4, 6).

Comment: 4. As claimed in the main text, only the size of atomic/ionic radii could not be confirmed as a factor for the increased or decreased lattice parameters. The bonding/anti-bonding and other properties could play vital role. I suggest authors to check this and amend accordingly.

Response:

We agree with the referee that the lattice parameters can be influenced by many factors besides the atomic/ionic radii. In our previous manuscript, we mainly utilized the atomic size to describe the trend of lattice constant, which fits well with Vegard’s law. The bonding and anti-bonding states in molecular orbitals also influence the bond lengths, according to a phenomenological bond length – bond strength rule. Short bonds are typically stronger bonds, while longer bonds are typically weaker bonds. Electrons in the bonding orbitals stabilize the molecule, pulling the nuclei closer and shortening the bond length. On the contrary, electrons in the anti-bonding states can destabilize the molecule by exerting a repulsive force between the nuclei. Thus, the bond length increases with increasing the population of occupied anti-bonding states. However, in this work, the occupied anti-bonding states are created by the overlap between Sn-s/dopant-s and Te-p orbitals. The destructive effect of these anti-bonding states is compensated for by the equivalent constructive effect of the bonding states formed by these orbitals. This has been proven and discussed in Fig. 1d. Thus, the anti-bonding state in SnTe has a negligible effect on the bond length. For GeTe, which closely resembles SnTe in terms of structure and bonding, the close relationship between

atomic arrangement and bonding has been discussed in quite some detail.¹ It has been shown that an increase in the Peierls distortion, which leads to a shortening of the short bond is accompanied by an increase in bond strength as quantified by the number of electrons shared between adjacent atoms. In this work, introducing the dopants in SnTe does not change the chemical bonding mechanism. The whole system remains metavalent bonding as verified by the high PME value determined by APT (Fig. 3). Thus, it appears that the atomic/ionic radii are the prominent factors for the variations in lattice parameters.

Revisions: We have added the discussions about the influence of bonding and anti-bonding states on the lattice parameters on page 13.

Reviewer #2:

Comment: In this work the thermoelectric performance of SnTe has been improved by co-doping Al and Sb and further alloying AgBiTe₂. The work focuses on the electronic structure tuning with an aim to increase the DOS effective mass. The work is good but needs to be revised adding additional data. Below are some of the points to be considered.

Response:

Thank you for your careful review and the insightful suggestions offered to improve our manuscript. We have provided a point-by-point response to your comments below. We hope the revisions made to the manuscript adequately address your concerns and significantly improve the quality of this work.

Comment: 1. The introduction should give a brief history of various dopants used to improve the DOS effective mass like V, W, Mo. Not only indium but there are other dopants like Bi, Zn which have the potential to introduce resonant states in SnTe. These dopants have versatile effects leading to increase in band gap, hyperconvergence, Rashba effect, introduction of multiple electronic valleys etc not only in SnTe but also in other telluride systems. Details on these need to be mentioned as electronic structure engineering is the approach adopted in this paper.

Response:

Thank you for providing this helpful comment to develop a more comprehensive overview of the electronic band structure engineering in both p-type and potential n-type SnTe. We have found that dopants such as V², W³, Mo⁴, and Zn^{5, 6} can also

introduce resonant states in SnTe besides the representative example of In. In particular, the effect of Zn on enhancing the local DOS of SnTe has also been highlighted in Fig. 2, where the symbol size of Zn is comparable to that of Ga and Tl. This further verifies the usefulness of the concept developed in this work to identify effective dopants for enhancing DOS by maximizing the spatial and energetic overlap between the dopant-s and Te-p orbitals.

Revisions: We have revised the “Introduction” part of the manuscript to include the important examples mentioned above. We have also highlighted the usefulness of Figure 2 in identifying effective dopants to enhance DOS by referring to the experimental results of In- and Zn-doped SnTe on page 12.

Comment: 2. The authors write “We have previously demonstrated that Sb doping produces negligible changes to the electronic structure of SnTe [41], which has also been shown and interpreted in Figure 2.” This is not accurate as doping of Sb has shown to introduce additional energy levels in the band gap region in SnTe. The authors need to modify the statement. Electronic structure of Sb doped SnTe should be provided. The authors mention the term “We” which previous work are the authors referring to?

Response:

Thank you for your suggestion. We have provided the electronic band structure of Sb-doped SnTe using a supercell corresponding to the experimental concentration of Sb (~15%) in Fig. R3. From the density of states of $\text{Sn}_{0.852}\text{Sb}_{0.148}\text{Te}$, we see a clear band gap without any additional energy levels in the band gap. This phenomenon has also been reported in previous literature. For example, Tan et al. have concluded that “*If the s-level of a dopant is lower than that of Sn/Pb-s, the dopant will also produce a molecular antibonding state, while its strength is weaker than that of in bulk. So, the whole antibonding state of the doped system will be weaker than the undoped system, but such decrease is very limited and only exhibits as weak BC (here BC means band convergence) behavior.*”⁷ We have also copied the calculated band structure of 7.4% Sb-doped SnTe from Ref⁷ as shown in Fig. R4. As has been concluded by Tan et al. “*The valence band is slightly tuned. In principle, we may not observe significant BC effects in Sb-doped system.*”⁷ Ho Seong Lee et al. claimed that “*both Bi and Sb have been classified by X. Tan et al. to have weak band convergence effects.*”⁸ Indeed, this observation is also consistent with our results and can be explained in Fig. 2. Due to the lower atomic energy and the smaller orbital radius of Sb-s states than that of Sn-s states, the overlap between Sb-s and Te-p is weaker than that of Sn-s and Te-p. Consequently,

Sb can slightly decrease the population of antibonding states at the L point and, thus, lower the energy at the L point to move the valence band maximum downward. This will lead to a weak band convergence effect, as reported by Tan et al. (see Fig. R4). Nevertheless, we indeed did not observe Sb-related gap states in SnTe.

Regarding the term “We” in the previous manuscript, there was a mistake in citing references. The correct reference should be our previous work published in *Advanced Functional Materials*, which shows that Sb doping has a relatively weak impact on the electronic structure of SnTe⁹. This conclusion is further confirmed in the present work by calculating the DOS of Sn_{0.852}Sb_{0.148}Te and understanding the spatial and energetic overlaps between dopant-s and Te-p orbitals, as illustrated in Fig. 2.

Fig. R3. Electronic DOS of pristine SnTe and Sn_{0.852}Sb_{0.148}Te. The Fermi energy level of SnTe is set at 0 eV, while the dashed line indicates the Fermi energy level of Sb-doped SnTe.

[Figure redacted]

Fig. R4. Electronic band structures of pure SnTe and Sb-doped SnTe. The figure is

copied from Ref⁷.

Revisions: We have added Fig. R3 as Fig. S6 to the supporting information. We have also corrected the reference on page 16. The literature work of Tan et al. has also been cited to show the consistency of the results obtained.

Comment: 3. Electronic structures for Al doped SnTe, Al and Sb co-doped SnTe and Al and Sb co-doped SnTe with AgBiTe₂ alloyed composition should also be provided.

Response:

Following your suggestions, we present the electronic structures of Al-doped, Al-Sb co-doped and Al-AgBiTe₂ doped SnTe in Fig. R5. The concentrations of Al, Sb, Ag and Bi are ~8%, ~15%, ~4% and ~4%, respectively. Due to the highly complex atomic arrangements in the six-element Ag-Sb-AgBiTe₂ doped SnTe system, we could not identify an ‘ideal’ random doping structure and thus could not obtain its electronic structure. Yet, it appears reasonable to remove Sb to simplify the DFT calculations since the modest role of Sb in the band structure of SnTe has been demonstrated above. Fig. R5a and its corresponding DOS in Fig. R5b clearly show an enhanced electronic state and a hump in DOS near the Fermi level. This has been thoroughly discussed in the manuscript. With further introducing Sb into the Al-doped SnTe, we do not observe significant changes in the valence band structure compared to that of Al-doped SnTe. The introduction of Sb mainly influences the conduction band with a strong contribution from the Sb-p orbital (Fig. R5c and d). Alloying AgBiTe₂ with the Al and Sb co-doped SnTe can not only retain the contribution from Al but also lower the energy offset ($\Delta E(L-\Sigma)$) between L and Σ points as shown in Fig. R5e-f. This illustrates that introducing Al and AgBiTe₂ in SnTe induces not only the stronger DOS peaks around the valence band maximum but also the smaller band offset which facilitates band convergence in SnTe. Both factors can increase the DOS effective mass and Seebeck coefficient.

Fig. R5. Electronic band structures of $\text{Sn}_{0.926}\text{Al}_{0.074}\text{Te}$, $\text{Sn}_{0.778}\text{Al}_{0.074}\text{Sb}_{0.148}\text{Te}$ and $\text{Sn}_{0.852}\text{Al}_{0.074}\text{Ag}_{0.037}\text{Bi}_{0.037}\text{Te}$.

Revisions: We have added Fig. R5 as Fig. S7 in the supporting information. We have made corresponding revisions to the manuscript on page 17 highlighted in yellow.

Comment: 4. Some of the samples show onset of bipolar conduction. Electronic structure should be used to explain this.

Response:

The onset temperature of bipolar conduction depends on both the electronic bandgap and the concentration of majority carriers. It is well-known that the bandgap calculated by DFT often deviates significantly from experimental values. In particular, the presence of in-gap states makes the determination of true bandgap more challenging. To avoid these potential problems, we measured the bandgap of $\text{Sn}_{1.03-x-y}\text{Al}_x\text{Sb}_y\text{Te}_z\text{AgBiTe}_2$ samples experimentally by UV-Vis-NIR absorption spectrum. The results are listed in Table R1. Even though pristine $\text{Sn}_{1.03}\text{Te}$ has a smaller band gap compared to other samples, it does not show bipolar conduction until 873 K (Fig. 4 in the manuscript) due to its high intrinsic carrier concentration. Doping Sb in SnTe can slightly increase the bandgap, as has been explained by Tan et al.¹⁰ and illustrated in Fig. R4. On the contrary, doping Al in SnTe should decrease the bandgap due to the increased population of occupied anti-bonding states which lift the L point. As a result, the bandgap first increases from SnTe to the sample $x=0.02$, $y=0.04$, $z=0$ and then

slightly decreases to the sample $x=0.08$, $y=0.015$, $z=0$ due to the co-doping of Al and Sb. The band gap further decreases with increasing the alloying content of AgBiTe₂. This phenomenon has also been observed by Tan et al.¹⁰ which can be attributed to two reasons. First, the bandgap of AgBiTe₂ (~0.16 eV) is smaller than that of SnTe and Al-Sb co-doped SnTe. Second, the lattice constant decreases with increasing the content of AgBiTe₂. This will enlarge the band dispersion and thus shrink the bandgap. Fig. 4c indicates that the bipolar effect is prominent at high temperatures only in samples with a high content of AgBiTe₂. The decreased bandgap is partly responsible for this behavior, while the reduced carrier concentration (Fig. 4e) is another reason for the enhanced bipolar conduction.

Table R1. Bandgap of Sn_{1.03-x-y}Al_xSb_yTe-z%AgBiTe₂ samples measured by UV-Vis-NIR absorption spectrum.

Sn _{1.03-x-y} Al _x Sb _y Te-z%AgBiTe ₂ samples	Bandgap (eV)
$x=y=z=0$	0.228
$x=0.02$, $y=0.04$, $z=0$	0.298
$x=0.08$, $y=0.015$, $z=0$	0.283
$x=0.08$, $y=0.015$, $z=2$	0.279
$x=0.08$, $y=0.015$, $z=4$	0.271
$x=0.08$, $y=0.015$, $z=6$	0.243

Revisions: We have added Table R1 as Table S3 in the supporting information. We have added the corresponding discussions on page 19 and in the supporting information.

Comment: 5. Ultra low lattice thermal conductivity has been observed in Cu and Sb doped SnTe also which is slightly higher (0.39 W/mK) than the present value. This should be included in the bar diagram.

Response:

Thanks for your suggestions. We have added this value in the bar diagram in Fig. 5c of the revised manuscript (Fig. R6).

Fig. R6. Comparison of κ_L of Sn_{0.8}Al_{0.08}Sb_{0.15}Te-4%AgBiTe₂ with other reported SnTe systems¹¹⁻¹⁸.

Revisions: We have included the lattice thermal conductivity from Ref [ACS Sustainable Chem. Eng. 2022, 10, 1367-1372]¹⁷ in Fig. 5c and updated this figure in the revised manuscript.

Comment: 6. The authors write “The average ZT (ZT_{ave}) of 1.28 is achieved for Sn_{0.8}Al_{0.08}Sb_{0.15}Te-4%AgBiTe₂ in the range from 400 K to 873 K (Figure 7b), which is a record-high value for SnTe systems^{22, 41, 55-58, 71.}” There are reports which show higher average ZT values than the ones cited here. The authors need to do a thorough literature survey and compare values with the highest values reported by far.

Response:

We sincerely appreciate the valuable suggestions. We have tried our best to carry out a thorough literature survey during the revision. Yet, we might miss some important papers due to the vast number of publications about SnTe thermoelectrics. The average ZT values between 300 and 873 K taken from literature and this work are compared in Fig. R7. The value obtained in this work is indeed the highest according to our knowledge. This large average ZT is mainly due to the significant improvement of ZT at low to intermediate temperature ranges although the maximum ZT of our sample at high temperatures is not the highest (please see the comparisons of maximum ZT in Fig. S15).

Fig. R7. Comparison of ZT_{ave} of $\text{Sn}_{0.8}\text{Al}_{0.08}\text{Sb}_{0.15}\text{Te}-4\%\text{AgBiTe}_2$ with other reported SnTe systems ^{5, 11-13, 15, 18-23}.

Revisions: We have re-calculated the average ZT between 300 and 873 K rather than between 400 and 873 K. More literature data have been added to Fig. 7b in the revised manuscript. The corresponding changes in text have been highlighted in yellow on page 26.

Comment: 7. Why is the value of ZT at 373 K highlighted in the abstract and conclusion? It is better to indicate the value at 300 K which is the lowest temperature of the study. Similarly for calculating average ZT the authors should consider 300 K instead of 400 K.

Response:

Thank you for your comments. We have provided the ZT value at 300 K in the revised version. ZT of 0.36 is achieved at 300 K, which is the highest value reported so far, as shown in Fig. R8. The average ZT has also been re-calculated between 300 K and 873 K, as shown in Fig. 7b (Fig. R7).

Fig. R8. Comparison of ZT (300 K) of $\text{Sn}_{0.8}\text{Al}_{0.08}\text{Sb}_{0.15}\text{Te}-4\%\text{AgBiTe}_2$ with other reported SnTe systems^{5, 9, 14, 15, 20-22, 24-26}.

Revisions: Fig. R8 has been included as Fig. S14 in the Supporting Information. The corresponding changes in text have been highlighted in yellow on page 26.

Comment: 8. It is better to update the references to include the ones published in the last 5 years to indicate the recent developments.

Response:

Thank you for your valuable suggestions. We have updated the references in the revised manuscript.

Reviewer #3:

Comment: The authors report the enhancement of ZT in SnTe by doping Al and Sb and alloying it with AgBiTe_2 . The rationale of the work is good but it can be improved further by considering the points below.

Response:

Thank you for your careful review and insightful suggestions offered to improve our manuscript.

Comment: 1. The introduction concentrates more on the carrier tuning than the electronic structure part. To give a clear state of art in the field of SnTe thermoelectrics it is essential to briefly mention various approaches adopted to improve the ZT by tuning the electronic structure like perfect convergence of light and heavy hole valence bands and light and heavy electron conduction bands, hyperconvergence of valence

bands and conduction bands, Rashba splitting, band gap increase etc. attained by doping.

Response:

We have followed your suggestions and have implemented more discussions on the electronic band structure engineering in SnTe. The convergence of light and heavy valence bands has been achieved in SnTe by introducing dopants such as Mg²⁷, Ca²⁸, Cd²⁹, Hg¹², and Mn³⁰. Besides the band convergence, introducing “resonant levels” is another important method to enhance DOS. We have added more examples that can introduce “resonant levels” in SnTe such as V², W³, Mo³, and Zn^{5, 6}. These dopants have also been reported to induce additional effects such as Rashba splitting and bandgap opening, which further tailor the electrical transport properties of SnTe.

Revisions: We have added the above discussions to the revised manuscript in the Introduction part.

Comment: 2. Various dopants and their impact on the electronic structure has to be provided in a tabular column.

Response:

Many thanks for your suggestions. We summarized various dopants and their impact on the electronic structure in Table S1 (Table R2).

Table R2. Thermoelectric transport properties of various dopants in SnTe-based materials. Seebeck coefficient (S) values at 300 K and maximum Seebeck coefficient, maximum power factor (PF), band effect, band convergence (BC) and resonant level (RL), electronic structure (bandgap calculated by DFT and energy offset between the light-hole band and heavy-hole band).

Dopants	S_{300K} ($\mu V K^{-1}$)	S_{max} ($\mu V K^{-1}$)	PF ($\mu W cm^{-1} K^{-2}$)	Band Effect	Band Gap (eV)	$\Delta E(L-\Sigma)$ (eV)	Ref
SnTe	24	110	15.7	/	0.06	0.35	This work
Ca	48	186	26	BC	0.07	0.2	28
Mg	38	200	30.3	BC	0.26	0.18	27
Mn	60	275	15.5	BC	0.08	0.2	14
Cd	51	200	19.1	BC	/	0.12	21
Bi-HgTe	60	178	24.2	BC	0.39	0.06	12
Ge-Sb	68	174	27	BC	/	0.15	24
Ge-Bi- AgBiTe ₂	60	190	25.8	BC	0.45	0.14	31
Ca-In	98	230	47	RL and BC	0.1 (Ca)	0.2(Ca)	32

AgInTe ₂	97	107	31.4	RL and BC	/	0.105	16
In	50	161	21.3	RL	/	/	18
Zn	127	205	42	RL	0.43	/	6
Bi-Zn	112	205	36	RL	/	0.27	5
Pb-Zn	109	229	30.4	RL	0.145	0.3	20
V	/	/	/	RL	/	/	2
W	/	/	/	RL	/	/	3
Al-Sb - AgBiTe ₂	106	190	28.11	RL and BC	0.271 (exp)	0.19	This work

Revisions: We have added Table R2 as Table S1 to the supporting information and referred to it in the main text on page 6.

Comment: 3. The authors need to clearly indicate what is the reason behind choosing AgBiTe₂ for alloying. As doping of Ag would have led to band convergence.

Response:

Thank you for raising this insightful question. Indeed, the reason for choosing AgBiTe₂ is closely related to the band convergence effect by doping Ag in SnTe. As stated by the referee, doping of Ag can lead to band convergence, which has been theoretically predicted³³ and experimentally proven³⁴. However, the solubility of Ag in SnTe is rather low (<1 at.%). This strongly limits the magnitude of band convergence. We have found that the solubility of dopants depends on the chemical bonding mechanisms of dopants and the host^{9,35}. Here, SnTe employs metavalent bonding. Thus, the solubility of Ag in SnTe will be significantly increased if it is doped in the form of a metavalently bonded compound such as AgSbTe₂ and AgBiTe₂. We have demonstrated the effectiveness of AgSbTe₂ in promoting band convergence in SnTe⁹. In this work, we chose AgBiTe₂ to increase the diversity of material choices to achieve band convergence.

Revisions: We have added a large paragraph to the Introduction part to explain the reason for choosing AgBiTe₂ for more effective band convergence.

Comment: 4. The electronic structure should be provided for the configuration showing maximum activity. That is SnTe doped with Al and Sb and Sn vacancies occupied by Ag and Bi.

Response: Following your suggestions, we have tried hard to calculate the electronic structure of the Al-Sb-Ag-Bi doped SnTe. Unfortunately, it is very difficult to identify

an ‘ideal’ random doping structure for the four-doping (Al/Sb/Ag/Bi)-SnTe compound within a supercell. Moreover, previous literature and our results (Fig. R3) have suggested that Sb has a weaker impact on the electronic structure of SnTe. In this regard, we calculated the electronic structures of $\text{Sn}_{0.882}\text{Al}_{0.074}\text{Ag}_{0.037}\text{Bi}_{0.037}\text{Te}$ to mimic the composition of the best-performing sample while leaving Sb out. From the electronic structures and the band offset ($\Delta E(\text{L}-\Sigma)$) between L and Σ points (Fig. S7 and Fig. S8), we notice that introducing Al/Ag/Bi in SnTe induces not only the strong DOS peaks around the valence band maximum but also the small band offset to enable band convergence in SnTe. These factors will facilitate the enhancement of DOS effective mass and Seebeck coefficients.

Revisions: Please find the corresponding revisions highlighted on page 17 in the revised manuscript.

Comment: 5. Alloyed samples show onset of bipolar conduction in Figure 4c. Correlation must be done with the electronic structure results.

Response:

The onset temperature of bipolar conduction depends on both the electronic bandgap and the concentration of majority carriers. It is well-known that the bandgap calculated by DFT often deviates significantly from experimental values. In particular, the presence of in-gap states makes the determination of true bandgap more challenging. To avoid these potential problems, we measured the bandgap of $\text{Sn}_{1.03-x-y}\text{Al}_x\text{Sb}_y\text{Te}_z\text{AgBiTe}_2$ samples experimentally by UV–Vis–NIR absorption spectrum. The results are listed in Table R1. Even though pristine $\text{Sn}_{1.03}\text{Te}$ has a smaller band gap compared to other samples, it does not show bipolar conduction until 873 K (Fig. 4 in the manuscript) due to its high intrinsic carrier concentration. Doping Sb in SnTe can slightly increase the bandgap, as has been explained by Tan et al.⁷ and illustrated in Fig. R4. On the contrary, doping Al in SnTe should decrease the bandgap due to the increased population of occupied anti-bonding states which lift the L point. As a result, the bandgap first increases from SnTe to the sample $x=0.02$, $y=0.04$, $z=0$ and then slightly decreases to the sample $x=0.08$, $y=0.015$, $z=0$ due to the co-doping of Al and Sb. The band gap further decreases with increasing the alloying content of AgBiTe_2 . This phenomenon has also been observed by Tan et al.⁷ which can be attributed to two reasons. First, the bandgap of AgBiTe_2 (~0.16 eV) is smaller than that of SnTe and Al-Sb co-doped SnTe. Second, the lattice constant decreases with increasing the content of

AgBiTe₂. This will enlarge the band dispersion and thus shrink the bandgap. Fig. 4c indicates that the bipolar effect is prominent at high temperatures only in samples with a high content of AgBiTe₂. The decreased bandgap is partly responsible for this behavior, while the reduced carrier concentration (Fig. 4e) is another reason for the enhanced bipolar conduction.

Table R1. Bandgap of Sn_{1.03-x-y}Al_xSb_yTe-z%AgBiTe₂ samples measured by UV-Vis-NIR absorption spectrum.

Sn _{1.03-x-y} Al _x Sb _y Te-z%AgBiTe ₂ samples	Bandgap (eV)
x=y=z=0	0.228
x=0.02, y=0.04, z=0	0.298
x=0.08, y=0.015, z=0	0.283
x=0.08, y=0.015, z=2	0.279
x=0.08, y=0.015, z=4	0.271
x=0.08, y=0.015, z=6	0.243

Revisions: We have added Table R1 as Table S3 in the supporting information. We have added the corresponding discussions on page 19 and in the supporting information.

Comment: 6. The authors need to clearly indicate what is the amount of Ag and Bi which enter into the lattice site of the host and what is the amount which remains undoped.

Response:

This question is closely related to our responses to your Comment #3. Energy dispersive X-ray spectroscopy and atom probe tomography analyses for the sample Sn_{0.80}Al_{0.08}Sb_{0.15}Te-4%AgBiTe₂ both show a composition of Ag and Bi close to their nominal stoichiometry (2 at% with a total amount of all elements being 100%). Moreover, no precipitates have been observed in a larger field of view in Fig. S5 (copied here as Fig. R9) and Fig. 6. This indicates that all Ag and Bi atoms have entered into the lattice site. The high solubility of Ag and Bi in SnTe is ascribed to the same chemical bonding mechanism for SnTe and AgBiTe₂. Both compounds employ metavalent bonding.

Fig. R9. (a) Scanning electron microscope (SEM) image and fracture surface SEM image of $\text{Sn}_{0.80}\text{Al}_{0.08}\text{Sb}_{0.15}\text{Te}-4\%\text{AgBiTe}_2$ (c-h) Elemental mapping of $\text{Sn}_{0.80}\text{Al}_{0.08}\text{Sb}_{0.15}\text{Te}-4\%\text{AgBiTe}_2$ taken from the area in (a).

Revisions: We have indicated the high solubility of dopants in SnTe and attributed the mechanisms underpinning this phenomenon to the same metavalent bonding mechanism for SnTe and AgBiTe_2 on page 14.

Comment: 7. A comparison plot (bar diagram) of room temperature (300 K) ZT values should be provided with previous reports of high value.

Response:

Thank you for your comments. A comparison plot (bar diagram) of room temperature (300 K) ZT values has been provided and compared with previously reported values (Fig. R7).

Revisions: Fig. R7 has been added in the supporting information as Fig. S14. We have also revised the corresponding descriptions about room-temperature ZT and average ZT in the manuscript on pages 26 and 27.

Comment: 8. A comparison plot (bar diagram) of peak ZT values with previously reported high performing SnTe materials (ZT above 1.5) should be given.

Response:

Thank you for your comments. Comparisons of maximum ZT value are presented in

Fig. R10.

Fig. R10. Comparison of ZT_{max} of $Sn_{0.8}Al_{0.08}Sb_{0.15}Te-4\%AgBiTe_2$ with other reported SnTe systems^{5, 12, 18-23, 26, 34}.

Revisions: Fig. R10 has been included in Supporting Information as Fig. S15.

Comment: 9. Average ZT should be reported between 300 K and 873 K.

Response:

We reported the average ZT between 300 K and 873 K in the revised version, as shown in Fig. 7b. We accordingly revised the sentences on page 26 and in the abstract.

Comment: 10. Have the authors tested the mechanical properties of the sample?

Response:

Thank you for this valuable suggestion. Microhardness and nanoindentation measurements were performed on $Sn_{1.03}Te$ and high-performance $Sn_{0.8}Al_{0.08}Sb_{0.15}Te-4\%AgBiTe_2$. More detailed descriptions can be found below in the “Revisions”.

Fig. R11. The Vickers microhardness and hardness (H) of $\text{Sn}_{1.03}\text{Te}$ and $\text{Sn}_{0.8}\text{Al}_{0.08}\text{Sb}_{0.15}\text{Te}-4\%\text{AgBiTe}_2$.

Revision: We have added the mechanical properties and corresponding descriptions to the revised manuscript on page 28. “Excellent mechanical properties are indispensable for well-established thermoelectric materials. Microhardness and nanoindentation measurements were performed on $\text{Sn}_{1.03}\text{Te}$ and high-performance $\text{Sn}_{0.8}\text{Al}_{0.08}\text{Sb}_{0.15}\text{Te}-4\%\text{AgBiTe}_2$. Vickers hardness and hardness (H) of $\text{Sn}_{0.8}\text{Al}_{0.08}\text{Sb}_{0.15}\text{Te}-4\%\text{AgBiTe}_2$ are both improved over pristine $\text{Sn}_{1.03}\text{Te}$, as shown in Fig. S21. The Vickers hardness increases from 65.6 HV0.1 for $\text{Sn}_{1.03}\text{Te}$ to 170.7 HV0.1 for $\text{Sn}_{0.8}\text{Al}_{0.08}\text{Sb}_{0.15}\text{Te}-4\%\text{AgBiTe}_2$. Meanwhile, a high hardness value of 2.5 GPa is achieved in $\text{Sn}_{0.8}\text{Al}_{0.08}\text{Sb}_{0.15}\text{Te}-4\%\text{AgBiTe}_2$, which is 132% higher than pristine $\text{Sn}_{1.03}\text{Te}$. This significantly enhanced mechanical properties can be attributed to the observed high density of dislocations. The enhancement of the mechanical properties of $\text{Sn}_{0.8}\text{Al}_{0.08}\text{Sb}_{0.15}\text{Te}-4\%\text{AgBiTe}_2$ enhances the stability of the thermoelectric properties during preparation and application and is also beneficial for thermoelectric devices to resist external mechanical shocks.”

Comment: 11. The authors need to take care of grammatical and spelling errors in the manuscript and supplementary information during revision.

Response:

Thank you for your suggestion. We have corrected grammatical and spelling errors during the revision stage.

References

1. Wuttig, M. *et al.* Revisiting the Nature of Chemical Bonding in Chalcogenides to Explain and Design their Properties. *Adv. Mater.* **35**, 2208485 (2023).
2. Shenoy, U. S., Bhat, D. K. Vanadium: a protean dopant in SnTe for augmenting its thermoelectric performance. *ACS Sustainable Chem. Eng.* **9**, 13033-13038 (2021).
3. Shenoy, U. S., Goutham, K. & Bhat, D. K. Resonance states and hyperconvergence induced by tungsten doping in SnTe: Multiband transport leading to a propitious thermoelectric material. *J. Alloy. Compd.* **905**, 164146 (2022).
4. Shenoy, U. S. & Bhat, D.K. Molybdenum as a versatile dopant in SnTe: a promising material for thermoelectric application. *Energy Advances* **1**, 9-14 (2022).
5. Shenoy, U. S. & Bhat, D.K. Bi and Zn co-doped SnTe thermoelectrics: interplay of resonance levels and heavy hole band dominance leading to enhanced performance and a record high room temperature ZT. *J. Mater. Chem. C* **8**, 2036-2042 (2020).
6. Bhat, D. & Shenoy, U. Zn: a versatile resonant dopant for SnTe thermoelectrics. *Mater. Today Phys.* **11**, 100158 (2019).
7. Tan, X. *et al.* Designing band engineering for thermoelectrics starting from the periodic table of elements. *Mater. Today Phys.* **7**, 35-44 (2018).
8. Kihoi, S.K. *et al.* Thermoelectric properties of Mn, Bi, and Sb co-doped SnTe with a low lattice thermal conductivity. *J. Alloy. Compd.* **806**, 361-369 (2019).
9. Liu, Y. *et al.* Improved solubility in metavalently bonded solid leads to band alignment, ultralow thermal conductivity, and high thermoelectric performance in SnTe. *Adv. Funct. Mater.* **32**, 2209980 (2022).
10. Tan, G. *et al.* SnTe–AgBiTe₂ as an efficient thermoelectric material with low thermal conductivity. *J. Mater. Chem. A* **2**, 20849-20854 (2014).
11. Banik, A., Vishal, B., Perumal, S., Datta, R. & Biswas, K. The origin of low thermal conductivity in Sn_{1-x}Sb_xTe: Phonon scattering via layered intergrowth nanostructures. *Energy Environ. Sci.* **9**, 2011-2019 (2016).
12. Tan, G. *et al.* Extraordinary role of Hg in enhancing the thermoelectric performance of p-type SnTe. *Energy Environ. Sci.* **8**, 267-277 (2015).
13. Wu, H. *et al.* Synergistically optimized electrical and thermal transport properties of SnTe via alloying high-solubility MnTe. *Energy Environ. Sci.* **8**, 3298-3312 (2015).
14. Li, W. *et al.* Promoting SnTe as an eco-friendly solution for p-PbTe thermoelectric via band convergence and interstitial defects. *Adv. Mater.* **29**, 1605887 (2017).
15. Hong, T. *et al.* Band convergence and nanostructure modulations lead to high

- thermoelectric performance in $\text{SnPb}_{0.04}\text{Te}-y\% \text{AgSbTe}_2$. *Mater. Today Phys.* **21**, 100505 (2021).
16. Banik, A., Shenoy, U. S., Saha, S., Waghmare, U. V. & Biswas, K. High power factor and enhanced thermoelectric performance of $\text{SnTe}-\text{AgInTe}_2$: synergistic effect of resonance level and valence band convergence. *J. Am. Chem. Soc.* **138**, 13068-13075 (2016).
 17. Kihoi, S. K. et al. Ultralow lattice thermal conductivity and enhanced mechanical properties of Cu and Sb Co-doped SnTe thermoelectric material with a complex microstructure evolution. *ACS Sustainable Chem. Eng.* **10**, 1367-1372 (2022).
 18. Zhang, Q., Liao, B., Lan, Y., Lukas, K. & Ren, Z. High thermoelectric performance by resonant dopant indium in nanostructured SnTe. *P. Natl. Acad. Sci. USA* **110**, 13261-13266 (2013).
 19. Guo, F. et al. Ultrahigh thermoelectric performance in environmentally friendly SnTe achieved through stress-induced lotus-seedpod-like grain boundaries. *Adv. Funct. Mater.* **31**, 2101554 (2021).
 20. Bhat, D. K. & Shenoy, U. S. SnTe thermoelectrics: dual step approach for enhanced performance. *J. Alloy. Compd.* **834**, 155181 (2020).
 21. Tan, G. et al. High thermoelectric performance of p-type SnTe via a synergistic band engineering and nanostructuring approach. *J. Am. Chem. Soc.* **136**, 7006-7017 (2014).
 22. Hong, T. et al. Realizing ultrahigh room-temperature seebeck coefficient and thermoelectric properties in SnTe-based alloys through carrier modulation and band convergence. *Acta Mater.* **261**, 119412 (2023).
 23. Tang, J. et al. Manipulation of band structure and interstitial defects for improving thermoelectric SnTe. *Adv. Funct. Mater.* **28**, 1803586 (2018).
 24. Banik, A. et al. Engineering ferroelectric instability to achieve ultralow thermal conductivity and high thermoelectric performance in $\text{Sn}_{1-x}\text{Ge}_x\text{Te}$. *Energy Environ. Sci.*
 25. Xu, X. et al. Constructing van der Waals gaps in cubic-structured SnTe-based thermoelectric materials. *Energy Environ. Sci.* **13**, 5135-5142 (2020).
 26. Chang, C. et al. Surface Functionalization of Surfactant-Free Particles: A Strategy to Tailor the Properties of Nanocomposites for Enhanced Thermoelectric Performance. *Angew. Chem. Int. Edit.* **61**, e202207002 (2022).
 27. Banik, A., Shenoy, U. S., Anand, S., Waghmare, U. V. & Biswas, K. Mg alloying in SnTe facilitates valence band convergence and optimizes thermoelectric properties. *Chem. Mater.* **27**, 581-587 (2015).

28. Al Rahal Al Orabi, R. *et al.* Band degeneracy, low thermal conductivity, and high thermoelectric figure of merit in SnTe–CaTe alloys. *Chem. Mater.* **28**, 376-384 (2016).
29. Tan, X. *et al.* Band engineering and improved thermoelectric performance in M-doped SnTe (M= Mg, Mn, Cd, and Hg). *Phys. Chem. Chem. Phys.* **18**, 7141-7147 (2016).
30. Wang, L. *et al.* Manipulating band convergence and resonant state in thermoelectric material SnTe by Mn–In codoping. *ACS Energy Lett.* **2**, 1203-1207 (2017).
31. Nie, C. *et al.* Band Modification and Localized Lattice Engineering Leads to High Thermoelectric performance in Ge and Bi Codoped SnTe-AgBiTe₂ Alloys. *Small* **19**, 2301298 (2023).
32. Bhat, D.K. & Shenoy, U. S. Enhanced thermoelectric performance of bulk tin telluride: Synergistic effect of calcium and indium co-doping. *Mater. Today Phys.* **4**, 12-18 (2018)
33. Lee, M. H., Byeon, D.-G., Rhyee, J.-S. & Ryu, B. Defect chemistry and enhancement of thermoelectric performance in Ag-doped Sn_{1+δ-x}Ag_xTe. *J. Mater. Chem. A* **5**, 2235-2242 (2017).
34. Sarkar, D. *et al.* Highly Converged Valence Bands and Ultralow Lattice Thermal Conductivity for High-Performance SnTe Thermoelectrics. *Angew. Chem. Int. Edit.* **59**, 11115-11122 (2020).
35. Liu, M. *et al.* Doping strategy in metavalently bonded materials for advancing thermoelectric performance. *Nat. Commun.* **15**, 8286 (2024).
<https://doi.org/10.1038/s41467-024-52645-3>.

Reviewer #1 (Remarks to the Author):

I found that the authors have adequately revised the manuscript addressing my comments. I recommend the article to be accepted for the publication.

Response: Thank you again for your constructive comments in the first review round and your recommendation for publication.

Reviewer #2 (Remarks to the Author):

The authors have thoroughly revised the manuscript. The manuscript in present form can be accepted for publication.

Response: Many thanks for your support.

Reviewer #3 (Remarks to the Author):

The authors have now revised the manuscript as suggested and the same may be accepted.

Response: Thank you for your kind suggestion and recommendation.

The manuscript entitled 'Interplay between metavalent bonds and dopant orbitals enables the design of SnTe thermoelectrics' presents a way of manipulating electronic structures to optimize the thermoelectric efficiency by doping Al and alloying AgBiTe₂ with SnTe. Overall, this work, with combined experimental and theoretical approach, looks extensive, well-organized, most of the relevant physical properties/parameters calculated, and results are promising for the TE device applications. So, I suggest the article to be published in this journal. However, I have following comments to be addressed in the revised manuscript before the acceptance and publication in this journal:

1. Figure 1c seems to be the key point to identify/prove metavalent bonding in SnTe. But it is not clear how the authors come up with the scheme in Figure 1c, i.e. how did SnTe show half-filled p-orbitals? I suggest adding more details/analyses on it.
2. About Figure 2, I suggest authors to add more detailed physical insights behind the usefulness of so called 'golden rectangle'. And what is/are the significance(s) of symbol size vs size of s-orbital of the dopants? Crucially, why the symbol size is scaled to difference between 0 eV and -0.1 eV, NOT -0.05 eV, -0.15 eV, ...? It could change things I believe.
3. In Figure S4a, y% refers to Sb element. However, in Figure S4b the plot shows y%, but the caption has Te-z% without y in the provided composition/compound. It is not clear.
4. As claimed in the main text, only the size of atomic/ionic radii could not be confirmed as a factor for the increased or decreased lattice parameters. The bonding/anti-bonding and other properties could play vital role. I suggest authors to check this and amend accordingly.